# Neuro-cognitive specificities in prosocial disobedience: A comparative fMRI study of civilian and military populations

Leslie Tricoche[1], Antonin Rovai[2], Salvatore Lo Bue[3], Emilie A. Caspar[1]*

1 Moral & Social Brain Lab, Department of Experimental Psychology, Ghent University, Belgium,
2 Translational Neuroanatomy and Neuroimaging Lab, Université Libre de Bruxelles, Belgium,
3 Department of Life Sciences, Royal Military Academy (RMA), Brussels, Belgium

* emilie.caspar@ugent.be

## Abstract

The literature highlighted that compliance with or resistance to authority orders to inflict pain involves cognitive processes like empathy, guilt, mentalization, cognitive conflict, and sense of agency. However, previous studies have focused on civilians, for whom such decisions are less significant than for military personnel, where obedience or resistance is integral to duty. This functional Magnetic Resonance Imaging (fMRI) study examined 53 military personnel, compared to 56 civilians, tasked with deciding whether or not to deliver shocks to a victim following orders received by an experimenter. Results revealed that military participants disobeyed orders less frequently, adopting higher homogeneous profiles: fewer ultra-prosocial and no antisocial behavior. High disobedience in military participants was driven by rational (moral, educational) and emotional factors (guilt, sadness, heightened victim sensitivity). Shared neuro-cognitive processes were observed between the two populations, but the level of engagement of these processes differed. While civilians predominantly recruited left temporo-parietal junction (TPJ) and dorso-medial prefrontal cortex (dmPFC) during decision-making, military agents relied more heavily on regions such as the right TPJ and anterior insula to achieve disobedience. Military participants also exhibited a reduced sense of agency in prosocial disobedience compared to civilians. In post-decision when witnessing the victim's pain, military participants preferentially activated right-hemisphere regions, while civilians engaged more left and medial regions. These differences suggest distinct mentalization forms, with civilians favoring cognitive empathy and military personnel relying more on affective empathy. These findings could have implications for ethics training in military and institutional contexts, and offer insights into how obedience and resistance are cognitively and emotionally constructed across different social roles and institutional environments.

**Data availability statement:** Data are made available on OSF (https://osf.io/xcgak/).

**Funding:** E.C.: ERC Starting Grant DISOBEY (grant number: 101075690) E.C.: BOF Starting Grant from Ghent University (grant number: BOF/STA/202109/025) E.C.: BIAL foundation (grant number: 150/18)

**Competing interests:** The authors have declared that no competing interests exist.

## Introduction

What drives a person to obey, or conversely, to resist an order, especially when that order is considered immoral because it harms another person? While this question has been explored in recent years through neuroimaging techniques such as functional Magnetic Resonance Imaging (fMRI), electroencephalography (EEG), and transcranial direct-current stimulation (tDCS) [1–3], the majority of the existing research have involved civilian participants—most often university students representing classic convenience samples, raising concerns about the generalizability of the findings (see for instance [1,2,4,5]). For these individuals, obedience may carry less existential and professional significance than for groups such as military personnel, whose daily lives are embedded in hierarchical structures and governed by duty, regulations, and the expectation to follow orders.

However, investigating these questions within military populations is a crucial societal and scientific endeavor, as both historical and contemporary events demonstrate that obedience to authority can lead to tragic outcomes. For instance, the Brereton Report (2020) investigated war crimes committed by Australian soldiers, and reported incidents in which junior soldiers were ordered to execute prisoners of war as part of a "blooding" ritual (Afghanistan Inquiry). This example highlights the profound human consequences of uncritical compliance and underscores the urgent need to understand how and when individuals—particularly those in military contexts—choose to resist immoral orders. Yet, military personnel, who are regularly required to carry out commands, must also be capable of evaluating the legitimacy of those orders and disobeying when necessary, in accordance with international humanitarian law (United Nations, 1950). Thus, resistance to authority is not only a psychological phenomenon but also a matter of critical societal importance—especially in contexts where following orders is routine.

Understanding how military personnel transition between compliance and resistance, as well as how their daily environment impacts these decisions, has not been approached in current literature. The present pre-registered study addresses this gap by investigating the neuro-cognitive mechanisms involved in resistance to immoral orders (i.e., prosocial disobedience) among military officer cadets and comparing them to civilians. Our goal is twofold: (1) to identify whether the same brain mechanisms observed in civilian populations are recruited when military participants choose to resist harmful orders, and (2) to determine whether the degree or nature of this recruitment differs across populations.

Prior studies in civilian adult participants have shown that compliance with or resistance to orders to inflict pain to another person involves cognitive and affective processes such as empathy [2,6–9], guilt [2,7,8], mentalizing [2,6], cognitive conflict [2,10], and sense of agency (SoA; defined as the subjective experience of controlling one's actions) [2,6,8,9,11–13]. These processes are engaged at different moments of the decision-making timeline [14]: from the initial processing of an order given by the experimenter (i.e., pre-decision phase), to the act of deciding to obey or disobey (i.e., decision-making phase), to the evaluation of its consequences for the victim (i.e., post-decision phase).

Notably, during post-decision, when witnessing the victim's pain, reduced activity in empathy for pain and guilt-related brain networks was observed when civilian participants complied with orders to cause pain compared to acting freely [7,9]. This result may help explain how obedience reduces our natural aversion to inflicting harm on others. Moreover, during the decision-making phase, studies in civilian participants revealed at both behavioral and brain levels that following orders led to a reduction of cognitive conflict and diminished SoA, which may facilitate the adoption of a non-moral behavior [8–13,15,16]. Interestingly, one previous work [13] showed similar neural patterns related to the SoA between civilians and military officers. However, these studies did not focus on disobedience *per se* in their research questions, primarily because adult participants almost never disobeyed orders in these experimental paradigms, making it impossible to compare the two types of decisions.

The literature on prosocial disobedience remains relatively limited but reveals consistent engagement of social and moral cognition networks, pointing to the emotional and physiological cost of resisting orders [3,4]. Specifically, a recent fMRI study with university students representing civilians found that during the decision-making period, up until the action, regions associated with moral judgment—specifically, the angular gyrus extending to the inferior occipital gyrus (AG/IOG), as well as the TPJ [17,18]—were negatively modulated in the disobedience condition compared to obedience [2]. Increasing activity in this region has previously been linked to a reduced SoA [19], suggesting that a negative modulation of the AG during decision-making could help maintain a sense of responsibility, thereby enabling resistance to the experimenter's order. Additionally, correlational analyses revealed a positive relationship between the rate of prosocial disobedience and the activity of medial regions when the decision to send a shock was made, particularly in the dorsomedial and ventromedial prefrontal cortex (dmPFC/vmPFC), the ACC, and the supplementary motor area (SMA) [2]. These regions are extensively involved in the process of mentalizing, including the distinction between self and others, and are therefore crucial in complex social situations that require different perspectives to be taken into account [20–22]. In the post-decision phase, Tricoche, Rovai and Caspar (2024) also found a positive correlation between the tendency to disobey and number of mentalizing regions over the fronto-parietal axis, more activated when the victim received a shock than a no shock [2]. It included mPFC, SMA, superior parietal lobule (SPL), supramarginal gyrus (SMG), TPJ, AG, precuneus (Prec), PCC and anterior insula (AI). These findings suggested heightened sensitivity to the consequences of their decisions, ultimately helping civilians to disobey more by improving the victim's perspective. Further support for the importance of mentalizing regions in prosocial disobedience comes from neurostimulation studies. For instance, cathodal stimulation of the right TPJ in Chinese adults, in a virtual replication of Milgram's paradigm, led to shortened reaction times to obey to harm an avatar, suggesting that disrupting other's perspective-taking processing can reduce resistance to immoral orders [1].

Despite these initial neural findings, the literature on prosocial disobedience remains limited and largely confined to civilian samples. The present study aims to expand this work by examining a group for whom obedience and disobedience are not only routine but institutionally prescribed—namely, military personnel. It specifically focused on prosocial disobedience, defined here as the refusal to comply with orders to inflict harm on another person. Indeed, it is important to distinguish between general disobedience and resistance to immoral orders specifically aimed at protecting others—as resistance rooted in moral concern could engage different mechanisms than disobedience driven by self-interest or norm compliance. Military officer cadets represent a particularly relevant population, as they are both trained to follow orders and educated in an ethical framework.

In the present fMRI study, military officer cadets, acting as agent, were instructed by an experimenter—on a trial-by-trial basis—to either deliver or withhold an electric shock to a victim's hand by pressing one of two buttons. The decision to obey or disobey these orders, by selecting the corresponding button, was entirely their own. This procedure was based on the paradigm of Caspar (2021) but adapted for fMRI [23]. We measured how often they resisted the order to send a shock (i.e., prosocial disobedience) and what were the associated neuro-cognitive processes. Their behavioral and brain-level results were compared with a previously published civilian dataset [2] using the same paradigm and fMRI

scanner. Almost no studies have compared civilian to military participants. However, since military officer cadets are more accustomed than civilians to receiving orders from authority figures, we hypothesized that they would show lower levels of disobedience. The reasoning behind their decisions might also differ due to their distinct educational environment and varying relationship with obedience. Moreover, two previous studies showed that SoA was the same among military officer cadets and civilians [8,13], and that it involved the same neural mechanisms [13]. We could thus expect similar brain regions involved during the decision phase among the two populations. However, as disobedience is less endorsed in military officer cadets and could pose a greater psychological challenge in the context of their professional identity, the involvement of this neural activity could differ. For other processes in the pre-decision and post-decision phase (i.e., cognitive conflict, empathy, guilt), our hypotheses remain bidirectional, as no previous studies exist with our target population.

## Materials and methods

### Participants

This experimental study was preregistered on OSF before data acquisition (https://osf.io/3mt4d). The study was approved in 2022 by the local ethics committee of Erasme Hospital (Belgium, project reference: SRB2022127), and it guaranteed the anonymity of participants during and after data collection. Using GPower and R, a priori computation of sample size indicated that we had to recruit a total sample of 1) 26 participants with ANOVAs, 2) 75 participants with correlations, and 3) at least 80 participants with segmented regressions; to achieve a power of.85, with a medium effect size (f = 0.25, r = 0.3, f² = 0.15) and a Type I error of 5%. To prevent unexpected data loss, we increased the sample size to at least 50 participants per group (i.e., a total of N ≥ 100) to ensure sufficient power for all our analyses. Fifty-four military officer cadets (3rd and 4th years students) were contacted and recruited through the Royal Military Academy of Belgium (22 identified as females, age range: 19−29 years, mean: 21.6 +/-1.8 years). All participants agreed to take part in the study between February 8, 2023, and April 23, 2024, and completed the fMRI session in which they played the role of an agent. Their results were compared with an equivalent sample of 56 civilian participants living in Belgium (33 identified as females, age range: 19−36 years, mean: 23.7 +/-4.2 years), who were recruited via social medias and previously analyzed (inclusion period: December 20, 2022 – May 15, 2023; under the same ethic approval), as reported in Tricoche, Rovai and Caspar, 2024 [2]. Participants had to meet the following inclusion criteria: being French, Dutch or English speakers; having normal or corrected vision; having no MRI contra-indication; and having no known neurological or psychological disorder. However, one military officer cadet was discarded from the fMRI analyses after discovering a significant anatomical abnormality during the scanning session that could compromise interpretation of imaging results and comparability across participants. Two other military officer cadets were discarded from the fMRI analyses due to a technical problem during the MRI-data acquisition, which resulted in a complete absence of usable MRI data. It led to a final sample size of N = 50 military officer cadets for imaging analyses. Concerning behavioral analyses, for one military officer cadet, only the Empathy run was analyzed due to a loss of data during the Agency run; while another one only completed the Agency run during the MRI session. A priori exclusion criteria also involved not being able to discriminate the different action-tone time intervals in the Agency run. We conducted a linear trend analysis with contrasts −1 0 1 to ensure that military officer cadets correctly reported shorter interval estimates (IE) for short delays and longer IE for longer action-outcomes delays. Five military officer cadets did not follow a linear trend, leading to non-significant difference between the three delays, and were excluded from the analyses using IE scores. It led to a final sample size of N = 53 (but N = 48 for analyses related to IE scores) military officer cadets for each run included in behavioral analyses. All participants (both civilians and military officer cadets) signed a written informed consent, and received a financial compensation between €30 and €40. Because military officer cadets receive a monthly allowance from the government, they are not permitted to earn additional income during their working hours. Therefore, any money they earned during the study was donated to an association or a non-governmental organization of their choice.

## Procedure

The following methodology was identical for civilian participants, as detailed in Tricoche, Rovai & Caspar, 2024 [2].

Military officer cadets came at the MRI scanner alone, and were introduced to an unknown age- and gender-matched partner, who acted as a second participant but was in reality a confederate. The military officer cadet took the role of "agent" whereas the partner took the role of "victim" (Fig 1A). Military officers came dressed in civilian clothing, as they are forbidden to wear their military uniforms outside military institutions unless they have special authorization related to their duties. To make the military officer cadets believe that the other person was also a participant, the experimenter welcomed them at the same time in the waiting room. In addition, all the information was provided similarly to the two participants, and the confederate asked questions such as any naïve participants would do.

To target distinct neuro-cognitive processes, the study employed two separate MRI runs, but with the same experimental design. This approach was based on findings from a previous study, which demonstrated that when the outcome of an action occurs more than four seconds after the action itself, the influence of sense of agency on time perception diminishes when using the temporal binding method [24]. To differentiate brain activity related to motor execution from that associated with the victim's pain perception, it was necessary to use relatively long and variable delays between action and outcome, which made time estimates impractical. During the Agency run—designed to focus on the pre-decision, decision-making, and post-decision phases for post-effects—military officers completed a temporal binding task. In contrast, the Empathy run was aimed at assessing the post-decision phase for outcomes and post-effects, with military officer cadets rating the pain experienced by the victim on a scale.

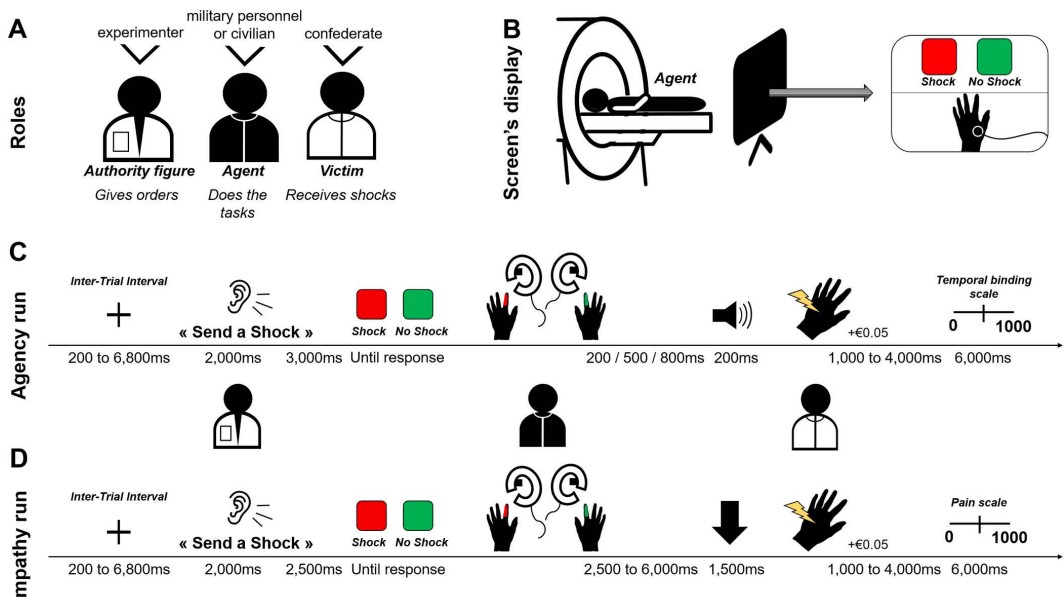

**Fig 1. Procedure, task and trial sequence.** Figure adapted from Tricoche, Rovai and Caspar, 2024 [2]. During the task, military officer cadets or civilian participants took the role of the agent, a confederate took the role of the victim, and the experimenter was associated to the authority figure (A). The screen seen by agents in the MRI scanner was split into two sections: the upper displayed the experimental interface, and the lower showed a live video feed of the victim's hand (B). Trial sequence of the Agency (C) and Empathy (D) runs. After receiving the auditory instruction ("Send a shock", "Do not send a shock"), agents must decide to obey or not to the experimenter order by pressing one of the two keys. Then, a tone occurred (for Agency) or an arrow was displayed (for Empathy). According to the outcome on the victim's hand (received shock or not), the agent could earn + €0.05. At the end of the trial, agents used the temporal binding scale (Agency) and the pain scale (Empathy).

After signing the consent forms, military officer cadets were individually trained for the two runs (Agency and Empathy) to ensure they understood the tasks. None of them required more than 12 trials to successfully complete the training. Then, the pain threshold was determined for both military officer cadet and confederate in front of each other, similar to the procedure used in Caspar et al. (2016) [12]. The electrodes were placed on the abductor pollicis muscle of the victim hand as it produced a visible muscle twitch, visible when a shock is delivered, in order to generate an empathic reaction when observing it [25,26]. The estimation of pain threshold's procedure ensured that military officer cadets knew that the shocks were real and how painful they were. Unknowingly to the military officer cadets, during the acquisition, the threshold of the victim was reduced up to a threshold that was not painful, but still triggered a visible muscle twitch. After the pain threshold procedure, the confederate stayed connected to the electric device, and the military officer cadet entered the scanner with a joystick in each hand. In front of the scanner there was a screen displaying both the experimental interface and the live video of the victim's hand (Fig 1B).

For both Agency and Empathy runs, on each trial, the military officer cadet was asked by the experimenter to send or not a shock to the victim ("Send a shock", "Do not send a shock"). The instructions were actually pre-recorded in 6 versions each, with small variations in the voice to control the timing of the audio instructions, but to let military officer cadets believe that the instructions were given in real-time. Military officer cadets had to decide to obey or not the experimenter's orders by pressing the corresponding "shock" or "no shock" button. To indicate which button was associated with shock and no shock on each trial, an image appeared, with a red square labelled "shock" and a green one labelled "no shock" displayed on the left and right parts of the screen. The mapping of the two buttons varied randomly to avoid response anticipation. Each time military officer cadets decided to give a shock, they earned +€0.05 in addition to their basic remuneration. For each run, the experimenter ordered military officer cadets to "send a shock" in 80 trials and to "not send a shock" in 40 trials, in a random order. Note that we offered military officer cadets the possibility of not following orders by explicitly mentioning this option in the instructions: "*I will ask you on each trial whether to send a shock to the victim or not. You will then choose to press one or the other button. Please know that you are not obliged to follow my instructions, I cannot force you to do anything, ultimately it is your decision*". This decision was made because, in pilot studies, we realized that participants never disobeyed in an MRI environment, thinking that they should not bias our study by doing so. Therefore, we introduced this small sentence to encourage military officer cadets to disobey more often, allowing for an analysis of disobedience.

## Task & trial sequence

Military officer cadets completed the Agency and Empathy runs in a counterbalanced order, in a single session.

For the Agency run, they had to decide, trial by trial, to obey or not to the experimenter order by pressing one of the two keys. Then, a 400 Hz-tone followed the keypress after a 200, 500 or 800ms pseudo-randomized delay. This tone occurred simultaneously with the outcome (i.e., shock or no shock delivered on the victim's hand). Military officer cadets had 6 seconds to estimate the action-outcome delay corresponding to the interval between the keypress and the tone [12], using a temporal binding scale from 0 to 1,000ms. The temporal binding task is an implicit method to measure SoA based on time perception [27,28]. More precisely, previous studies showed that performing voluntary actions led to a compression of time between the action and the outcome, compared to when the action was performed involuntarily (i.e., temporal binding effect) [29,30]. Trial sequence of the run is illustrated in Fig 1C.

For the Empathy run, after the military officer cadet's decision to obey or not the experimenter order, an arrow pointing to the video of the victim's hand was displayed during 1,500ms. This ensured that military officer cadets focused on the hand during the outcome (shock/no shock). The shock was delivered between 2.5 and 6s after the keypress to ensure no overlap between the motor response of the keypress and the processing of the victim's pain [6]. Then, a pain rating scale from 0 ("not painful at all) to 1,000 ("very painful"), appeared randomly in 20% of the trials (24/120 trials). Military officer cadets had 6 seconds to rate the intensity of the victim's pain, according to what they saw. Trial sequence of the run is illustrated in Fig 1D.

## Questionnaires

Military officer cadets completed several questionnaires before and after the MRI session and answered debriefing questions to investigate several individual and social dimensions, and their potential modulation on the results. Only the results concerning the debriefing questions are reported in the following results, while the used questionnaires and their analyses are available in S1 File. The debriefing questions particularly investigated 1) if military officer cadets voluntarily decided to disobey, 2) several subjective feelings by rating the explicit feeling of responsibility, feeling of being bad, feeling of being sad, and the perception of the shocks' painfulness, using scales ranging from "not at all" to "extremely" (e.g., "*How bad did you feel when you sent a shock to the victim?*"), and 3) why they obeyed or disobeyed orders, also by using scales from "not at all" to "extremely" (e.g., "*because there were too many shocks*", "*for moral reasons*", "*to win more money*", "*because it was the aim of the experiment*", …).

## Behavioral data and analyses

Analyses were conducted using R (RStudio, v.4.0.0). The same parameters as the ones described in Tricoche, Rovai and Caspar, 2024 [2] were used to ensure a full comparison between civilians and military officer cadets. For both runs, we measured the Prosocial Disobedience rate (%Pro_disob), corresponding to the number of times the agents refused to deliver a shock to the victim, despite the "Send a shock" order given by the experimenter.

Implicit Sense of Agency (SoA) was targeted by measuring the temporal binding score, similarly to previous studies [12,27,28]. During the Agency run military officer cadets were asked to estimate the duration (0–1,000ms) between their keypress (action) and the tone associated to the shock/no shock (outcome). IE were transformed into z-scores: $z-score = \frac{IE_i - mean\ IEs}{sd\ IEs}$, where $i$ corresponds to one trial. We also calculated the mean difference between the IE and the actual delay (error-score), as a second indicator of temporal binding. Then, higher IEs (i.e., positive z-score or positive error-score) are associated with a lower SoA; and reciprocally.

Empathy was estimated with a subjective pain scale (values between 0–1,000) rated by military officer cadets to report the intensity of the pain caused on the victim according to the received/not received shock.

For the Agency run, three-way Instruction ("Send a shock", "Do not send a Shock") X Choice (Obedience, Disobedience) X Population (Civilian, Military) ANOVAs were conducted on the mean temporal binding z-score and error-score. The same ANOVA was conducted on the mean pain score for the Empathy run. A four-way ANOVA adding the between-subject factor Run (Agency, Empathy) was conducted on the %Pro_disob. The assumptions of normality were assessed through visual inspection of the residuals using histograms and Q–Q plots. The distributions appeared reasonably symmetric and did not show substantial deviations from normality, excepted for the pain score. However, ANOVA was retained for this measure due to the large sample size and its robustness under such conditions [31]. Pairwise comparisons with False Discovery Rate (FDR) correction for multiple comparisons were conducted to interpret significant interactions [32]. Effect sizes ($\eta_p^2$) and confidence intervals at 95% ($CI_{95}$)are also reported.

Using Pearson correlation tests (or the equivalent non-parametric Spearman test when the normality criteria was not complied) we conducted analyses with correlations between the %Pro_disob and the mean temporal binding z-score or error-score, as well as the pain score. For this analysis we used the mean pain score obtained during the trials where military officer cadets obeyed to send a shock. The %Pro_disob was also correlated with the self-reported feelings sub-scales and the reasons of disobedience sub-scales completed during the debriefing phase. To investigate if the relation between the %Pro_disob and the behavioral (scale scores) or subjective measures (debriefing sub-scales) was modulated by the Population (Military, Civilians), we adjusted a segmented linear regression model to the data for each run ('segmented' package on R). This approach was chosen because it allows for the detection of nonlinearities or structural changes in the relationship between variables (%Pro_disob and Population)—particularly useful when interactions with categorical groups (such as Population) may result in different patterns. Indeed, unlike standard linear regression, which assumes a single, constant slope across the entire range of the independent variable, segmented regression allows the

model to estimate different slopes, informing when a relationship shifts at a certain threshold, depending on the population. The relation was investigated in the two directions leading to two models: 1) Measure~%Pro_disob x Population; 2) %Pro_disob~Measure x Population. This bidirectional modeling strategy enabled us to explore whether disobedience rates predicted variations in behavioral or subjective responses and conversely, whether individual differences influenced disobedience rates, across the two populations. It should be noted that these are complementary analyses that were not initially specified in the pre-registration.

Significance level was set at p < 0.05 for all analyses. To complement these frequentist statistics, we also conducted Bayesian analyses using JASP. For ANOVAS, we calculated the Bayes Factor inclusion ($BF_{incl}$) that compare all models including an effect to all models that do not include the effect. For correlation tests we calculated the Bayes Factor 10 ($BF_{10}$).

### fMRI data acquisition

MRI images were recorded using a research-dedicated hybrid 3.0-Tesla SIGNA Positron Emission Tomography-Magnetic Resonance (PET-MR) scanner (GE Healthcare, Milwaukee, Wisconsin, USA) and a 24-channel head and neck coil, at Erasme Hospital in Brussels (Belgium). Two runs, one for Agency and one for Empathy, of functional images were recorded in a single-shot echo planar imaging (EPI) sequence with the following parameters: matrix = 96x96; field-of-view (FOV) = 288 × 288 mm; number of slices = 45; TR = 2.2 seconds; TE = 20ms; flip angle = 90°; voxel size = 3x3x3mm³; slice thickness = 3 mm; acquisition order = interleaved (bottom-up)). The Agency functional run lasted approximately 30–35 minutes, and the Empathy run approximately 20–25 minutes, depending on the number of volumes acquired per participant. The time of the Agency Run was longer as participants also had to complete a time estimation task as a proxy for SoA. Between the two functional runs, T1-weighted 3D structural images were acquired (matrix = 240x192; TR = 8.2ms; TE = 3.1ms; flip angle = 12°; voxel size = 0.94x0.94x0.5mm³, slice thickness = 1 mm). A functional resting-state acquisition was added at the end of the experiment, but it was not analyzed in the present study. At the beginning of each functional run, four dummy scans were acquired to allow for magnetic field stabilization, and were automatically discarded by the scanner. Overall, the MRI session lasted approximately one and a half hours per participant.

### fMRI data preprocessing and processing

MRI data were preprocessed using fMRIPrep 20.2.0 [33]. T1 images were segmented and normalized to the MNI space. Functional images were realigned, slice-time corrected, coregistered and warped to the normalized anatomical image (for a full report of the preprocessing pipeline see https://osf.io/xcgak/). Using SPM12 (Statistical Parametric Mapping software; SPM12), images were spatially smoothed with an 6x6x6 mm³ kernel.

At the first level, we defined separate regressors for "Send a shock" and "Do not send a shock" instructions and for Obedience and Disobedience choices, with Agency and Empathy trials modelled in separated runs. For the Agency run, activation was modeled as epochs with 3 distinct onset times. The first one started with the screen display of the "shock"/"no shock" key associated responses and lasted until the tone, in order to target the decision-making phase for action. In fact, this epoch corresponded to the period between the decision-making and the outcome, including the motor action. The second one was the 3,000ms delay after the auditory instruction, in order to target the pre-decision phase for auditory processing. Finally, the delay between the tone and the interval estimate scale display was used to target the post-decision phase for post-effects, as it was the period after the outcome [34]. As this delay was the same for the two runs, the post-effect period was investigated for both Agency and Empathy runs (i.e., between the arrow display and the pain scale display), and merged together in the analyses (controlling for the Run effect).

For the Empathy run, we also modeled the hemodynamic response using the onset time starting with the display of the arrow pointing to the video of the victim's hand, in order to target the post-decision phase for outcome [7].

A regressor of no interest included all the other events: (1) the auditory instructions, (2) the visual display of the temporal binding scale, (3) the visual display of the pain scales, and (4) the keypress and the tone (excepted for SoA in Agency run). The six motion regressors were also added in the model.

## fMRI data analysis

### MRI data analyses were performed using SPM12. Whole-brain analyses

Four contrasts of interest were used and applied on the general linear models (GLM) for each decision phase (pre-decision, decision-making, and post-decision for outcome or post-effects):

- The ["Send a shock" > "Do not send a shock"] contrast, and its reverse contrast, investigated the effect of the received auditory instruction during the pre-decision phase.
- The ["Send a shock"/Disobedience > "Send a shock"/Obedience] contrast, and its reverse contrast, investigated the neural signature of prosocial disobedience (i.e., same order but different decision). This contrast was particularly analyzed for the pre- and decision-making phases, as analyses on the epochs associated with the post-decision phase would be unreliable because they would contrast a trial with no shocks to a trial with a shock.
- The ["Send a shock"/Obedience > "Do not send a shock"/Obedience] contrast, and its reverse contrast, investigated obedience trials, depending on the orders received, for all decision phases.
- The ["Send a shock"/Obedience + "Do not send a shock"/Disobedience] > ["Do not send a shock"/Obedience + "Send a shock"/Disobedience] contrast, and its reverse contrast, investigated the victim's pain processing, by comparing trails where the victim received a shock *versus* no shock, during the decision-making and post-decision phases particularly. Even if military officer cadets did not already see the victim receiving or not a shock during the decision-making phase, they already made decision and knew the resulting outcome of their keypress. We thus included this phase in this contrast.

As the antisocial disobedience rates of military officer cadets were very low (i.e., hearing an order not to send a shock but still pressing the shock button, see Results section below), the contrasts requiring these decision types could not be analyzed, comprising the full model including the four decision types (i.e., "Send a shock"/Disobedience, "Send a shock"/Obedience, "Do not send a shock"/Disobedience, "Do not send a shock"/Obedience).

To ensure the reliability of the *f*MRI results emerging from these contrasts, and based on previous studies [2,7], we removed military officer cadets who did not have a sufficient number of trials (<5 trials) in one or several of the conditions considered for a given contrast.

To investigate differences between military officer cadets and civilians, we conducted analyses adding the Population between-subject factor for each contrast of interest. For all our contrasts, we used a significance threshold of $p < 0.05$ (FWE corrected for multiple comparisons) at the cluster level, with an initial voxel-wise probability threshold of $p < 0.001$ uncorrected.

## ROI analyses

We used the same 17 ROIs as used in Tricoche, Rovai and Caspar, 2024 [2] (where exact coordinates are given): bilateral anterior insula (AI), bilateral inferior occipital gyrus extending to angular gyrus (IOG/AG), bilateral precentral gyrus (PreG), bilateral supramarginal gryus (SMG), bilateral superior parietal lobule (SPL), bilateral temporo-parietal junction (TPJ), precuneus extending to posterior cingulate cortex (Prec/PCC), bilateral supplementary motor area (SMA), ventro-median prefrontal cortex extending to anterior cingulate cortex (vmPFC/ACC) and dorso-median prefrontal cortex (dmPFC). In each ROI (built as the intersection of 10 mm radius spheres centered on the local maximum of each cluster), the activity was averaged across all voxels. The mean beta values were extracted from both the ["Send a shock"/Obedience > "Send a shock"/Disobedience] (i.e., same order but different choices) and the ["Send a shock"/Obedience > "Do not send a shock"/Obedience] (i.e., different orders but similar choices) contrasts for each ROI. Spearman correlations

with FDR correction were conducted between the mean beta values and the %Pro_disob. Both frequentist and Bayesian analyses were conducted, and effect sizes (r) were reported.

Finally, to investigate if the relation between the %Pro_disob and the ROI's activity was modulated by the Population, we conducted a segmented regression model analysis. The relation was investigated in the two directions leading to two models: 1) ROI activity ~ %Pro_disob x Population; 2) %Pro_disob ~ ROI activity x Population. As an exploratory analysis, we also conducted segmented regressions between activity of ROIs showing significant difference between the two populations and the disobedience's criteria (Model 1: ROI activity ~ Criterion x Population; Model 2: Criterion ~ ROI activity x Population).

## Results

### Behavioral results

**Percentage of disobedience.** All military officer cadets refused to send a shock in at least one trial. Sixty-two percent (33/53) for the Agency run and 66% (35/53) for the Empathy run of military officer cadets disobeyed prosocially (i.e., %Pro_disob) in at least 10% of the trials. For comparison, these percentages were respectively 75% and 71% in civilians. Also, N = 6 military officer cadets adopted a high altruistic profile, with more than 90% of prosocial disobedience trials across the two runs (≥72/80 trials). Contrary to civilians where five agents extensively sent a shock even when the experiment did not order it (i.e., antisocial disobedience), no military officer cadets adopted this behavior (Fig 2A).

The Instruction x Choice x Run x Population ANOVA on the percentage of choice revealed a main effect of Instruction ($F_{(1,651)}=120.12$, $p<0.001$, $\eta_p^2=0.16$, $BF_{incl}>100$ in favor of H1), a main effect of Choice ($F_{(1,651)}=183.75$, $p<0.001$, $\eta_p^2=0.22$, $BF_{incl}>100$ in favor of H1), as well as a significant interaction between these two factors ($F_{(1,651)}=8.29$, $p=0.004$, $\eta_p^2=0.01$, $BF_{incl}>100$ in favor of H1). As the number of instructions to give a shock (i.e., 80/120 trials) was higher than the order to not give a shock (i.e., 40/120 trials), the main effect of instruction was expected (mean "Send a shock" = 35.9%, $CI_{95}: \pm 2.3\%$, mean "Do not send a shock" = 20.1%, $CI_{95}: \pm 1.4\%$). The main effect of Choice indicates that the obedience rate was significantly higher than the disobedience rate (mean Obedience = 35.8%, $CI_{95}: \pm 1.6\%$, mean Disobedience = 19.3%, $CI_{95}: \pm 2.3\%$). The significant interaction was due to a non-significant difference ($p=0.41$) between the two prosocial decisions: obedience to not send a shock (mean = 29.9%, $CI_{95}: \pm 0.8\%$,) and disobedience to send a shock (mean = 28.51%, $CI_{95}: \pm 3.2\%$), while the other conditions differed between each other (all $ps<0.001$). The main effect of Population ($F_{(1,97)}=38.91$, $p<0.001$, $\eta_p^2=0.29$, $BF_{incl}>100$ in favor of H1) and the Choice x Population interaction were also significant ($F_{(1,651)}=7.79$, $p=0.005$, $\eta_p^2=0.01$, $BF_{incl}>100$ in favor of H1). As showed by post-hoc comparisons, civilians disobeyed more compared to military officer cadets, both prosocially ($t=2.18$, df = 768, $p=0.04$) and antisocially ($t=2.03$, df = 768, $p=0.052$) (Fig 2B). We did not find a significant main effect of Run nor interactions with the other factors, suggesting that participants disobeyed at the same extent in the two runs (all $ps>0.40$, all $BF_{incl}s<0.03$ in favor of H0). No other interaction was significant nor supporting H1 (all $ps>0.50$, all $BF_{incl}s<3$). To ensure the robustness of our findings, we also computed the percentage of choices relative to the number of trials corresponding to each instruction ("Send a shock" = 80 trials; "Do not send a shock" = 40 trials). The results, reported in S2 File, were comparable to those presented here.

S2 File reports the obedience and disobedience rates for each military officer cadet (the rates for civilians are available in Tricoche, Rovai and Caspar, 2024 [2]). We also plotted individual curves showing cumulative prosocial disobedience across the 120 trials for each run to visualize prosocial disobedience over time (S1 Fig).

### Temporal binding (proxy for the sense of agency)

We conducted a repeated-measure ANOVA with two within-subject factors Instruction and Choice and the between-subject factor Population on IEs z-scores and error-scores as the dependent variable. For z-scores, we found a main effect of Choice ($F_{(1,256)}=11.52$, $p<0.001$, $\eta_p^2=0.04$, $BF_{incl}=1.87$), along with a main effect of Population ($F_{(1,97)}=4.12$,

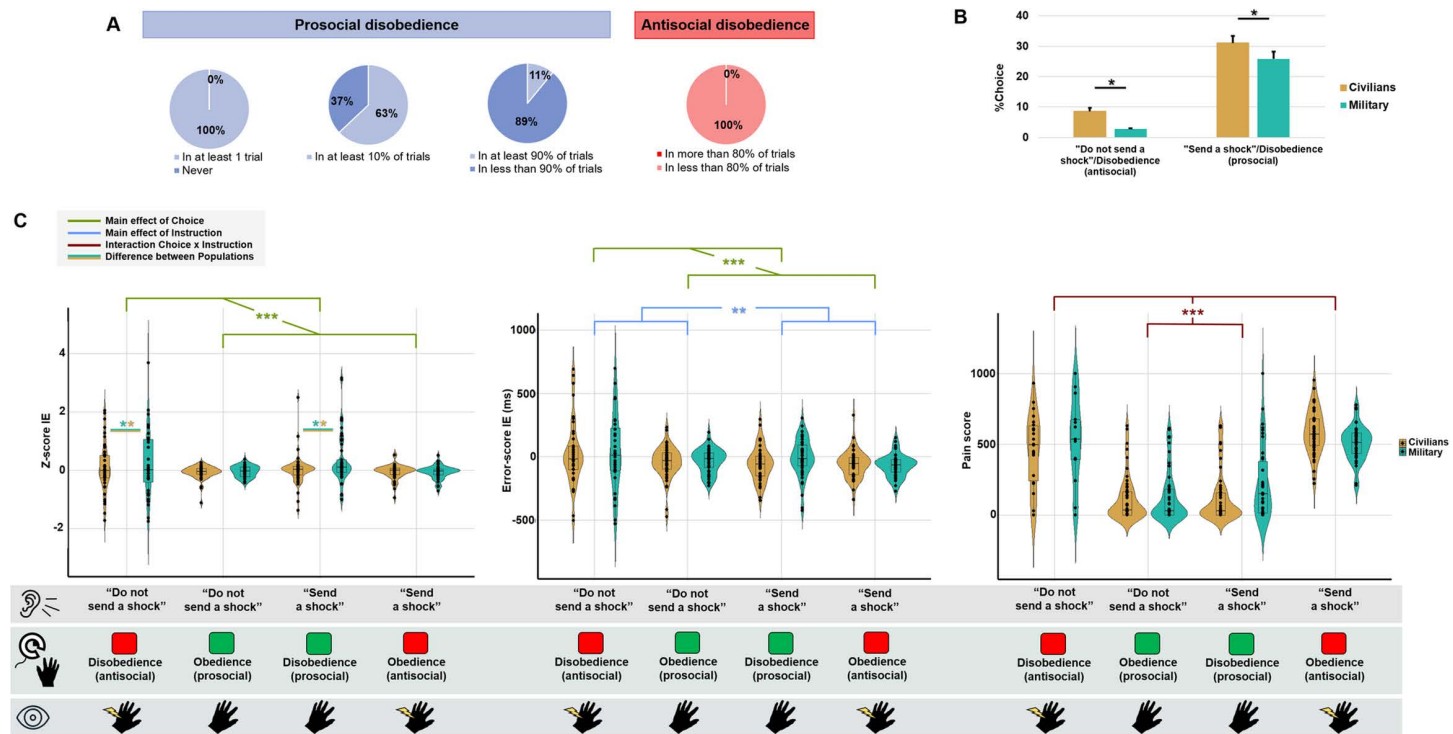

**Fig 2. Behavioral results.** (A) Almost all military officer cadets disobeyed but to a lesser extent compared to previous results obtained from civilians. Military officer cadets exhibited a more homogenous profile, with only a few showing highly prosocial and none demonstrating high levels of antisocial disobedience. (B) Bar plots show that military officer cadets disobeyed less than civilians, regardless of type of disobedience (prosocial or antisocial). (C) Box plots for Z-score, Error-score, and Pain score, showing the median and the interquartile range ± the standard deviation. Black dots represent individual values for each agent. Note that positive Z-scores and Error-scores (i.e., higher interval estimates) are associated with a lower SoA, and vice versa. Significant lines in green, blue and red indicate the main effect of Choice, the main effect of Instruction, and the Choice x Instruction interaction, respectively. The Z-score was higher for disobedience in military officer cadets compared to civilians. (** $p < 0.01$, *** $p < 0.001$).

$p = 0.04$, $\eta_p^2 = 0.04$, $BF_{incl} = 0.22$) and a Choice x Population interaction (F(1,256)=3.92, $p = 0.049$, $\eta_p^2 = 0.01$, $BF_{incl} = 0.33$). The z-score was higher, reflecting a lower SoA, for disobedience than obedience choices (mean Obedience = −0.06, $CI_{95}$: ±0.03, mean Disobedience = 0.16, $CI_{95}$: ±0.13), particularly for military officer cadets, as post-hoc analyses indicated significant differences between the two populations for disobedience only ($p = 0.01$) (Fig 2C). For error-scores, significant results were obtained only for the main effects of Choice (F(1,256)=10.66, $p = 0.001$, $\eta_p^2 = 0.04$, $BF_{incl} = 1.39$) and Instruction (F(1,256)=4.90, $p = 0.03$, $\eta_p^2 = 0.02$, $BF_{incl} = 0.25$). In line with the z-score, the error-score was higher, indicating less SoA, for disobedience than obedience (mean Obedience = −51.89, $CI_{95}$: ±14.87, mean Disobedience = −5.41, $CI_{95}$: ±31.77. Also, the instruction to shock led to higher SoA than when asking to not send a shock (mean "Send a shock" = −47.27, $CI_{95}$: ±17.11, mean "Do not send a shock" = −10.96, $CI_{95}$: ±30.29). However, for both z-cores and error-scores, Bayesian analyses led to inconclusive results or in support to H0. No difference was found between military officer cadets and civilians (all ps > 0.18, all $BF_{incl}s < 0.1$ in favor of H0) (Fig 2C).

## Subjective pain

We conducted the same repeated-measures ANOVA with Instruction, Choice and Population as factors on the pain scale. As expected, the main effects of Choice and Instruction were significant (Choice: F(1,219)=36.05, $p < 0.001$, $\eta_p^2 = 0.14$, $BF_{incl} > 100$; Instruction: F(1,219)=61.75, $p < 0.001$, $\eta_p^2 = 0.22$, $BF_{incl} > 100$), as well as their interaction (F(1,219)=280.19, $p < 0.001$, $\eta_p^2 = 0.56$, $BF_{incl} > 100$). All Bayes factors indicated strong evidence for H1. The conditions in which the victim

actually received a shock led to higher pain scores (ps<0.001; "Send a shock"/Disobedience: 168.38, $CI_{95}$: ±44.64; "Do not send a shock"/Obedience: 120.74, $CI_{95}$: ±32.88; "Send a shock"/Obedience: 544.57, $CI_{95}$: ±29.3; "Do not send a shock"/Disobedience: 475.08, $CI_{95}$: ±87.05) (Fig 2C).

The Choice x Population interaction was also significant (F(1,219)=11.37, p<0.001, $\eta_p^2$=0.05, $BF_{incl}$=11.94 in favor of H1). Post-hoc comparisons showed that the difference in pain scores between obedience and disobedience was only significant for civilians, who reported higher pain scores when obeying compared to disobeying orders (Civilians: t=2.73, df=330, p=0.04; Military: t=0.26, df=330, p=0.79).

## Relationship between prosocial disobedience and behavioral measures

In military officer cadets, significant correlations were found between %Pro_disob and IE scores (Z-score: r=−0.43, p=0.002, $BF_{10}$=11.48; Error-score: r=−0.29, p=0.048, $BF_{10}$=1.70 inconclusive). These negative correlations indicated that a lower interval estimate, reflecting higher SoA, was associated with a higher tendency to refuse to send a shock. For the pain score when obeying an order to send a shock, a marginal positive correlation emerged with the %Pro_disob (r=0.18, p=0.08, $BF_{10}$=0.6 in favor of H0), showing that a higher pain score when the military officer cadets witnessed a shock was associated with a higher tendency to refuse to send a shock. To explore differences in relationships between the two populations, we performed a segmented regression model analysis. No significant effect of Population was found on the relationship (all t's<|1.3|, all p's>0.2), suggesting that the association between %Pro_disob and behavioral measures (IE scores, pain score) was consistent across both populations (Fig 3).

## Qualitative results

We investigated the correlations between %Pro_disob (for Agency and Empathy runs separately) and the military officer cadets' subjective feelings, as well as their self-reported reasons for disobeying, rated during the debriefing phase. Significant positive correlations with mainly strong evidence toward H1, were found in both runs across all three emotional measures (Agency: feel responsible: r=0.29, $p_{FDR}$=0.03, $BF_{10}$=2.32; feel bad: r=0.47, $p_{FDR}$<0.001, $BF_{10}$>100; feel sorry: r=0.54, $p_{FDR}$<0.001, $BF_{10}$>100; Empathy: feel responsible: r=0.31, $p_{FDR}$<0.001, $BF_{10}$=0.79; feel bad: r=0.59, $p_{FDR}$<0.001, $BF_{10}$>100; feel sorry: r=0.65, $p_{FDR}$<0.001, $BF_{10}$>100), as well as for the perception of the shocks' painfulness (Agency: r=0.36, $p_{FDR}$=0.01, $BF_{10}$=7.36; Empathy: r=0.34, $p_{FDR}$<0.001, $BF_{10}$=4.66). A greater tendency to disobey was

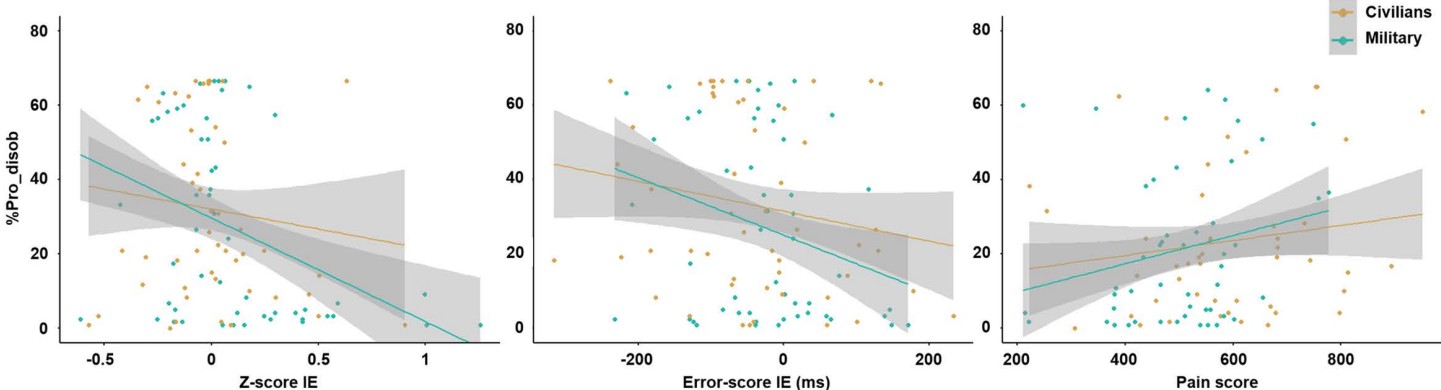

**Fig 3. Segmented regression model analysis.** This analysis shows the relationship between the scale scores (Z-score IE, Error-score IE, Pain score) and %Pro_disob, accounting for the Population factor (Civilians, Military). No Population effect was found, suggesting that the relationships between %Pro_disob and all scores were consistent across both populations (i.e.,a negative relationship between IE scores and %Pro_disob and a positive relationship between pain score and %Pro_disob).

associated with stronger feelings of responsibility, heightened guilt, increased sadness, and greater sensitivity to the shocks' perceived pain.

Regarding the reasons for disobeying (i.e., morality, sensitivity, shock quantity, money, entertainment, rejection of authority, experimental, fear of judgment, education, and country's history criterions), we found positive correlations with evidence toward H1, between the %Pro_disob and the criterion of morality ("*How much did you disobey for moral reasons?*") (Agency: r=0.62, $p_{FDR}$<0.001, $BF_{10}$>100; Empathy: r=0.55, $p_{FDR}$=0.002, $BF_{10}$=66.61), sensitivity ("*How much did feeling bad for the victim influence your decision?*") (Agency: r=0.48, $p_{FDR}$=0.005, $BF_{10}$=47.37; Empathy: r=0.44, $p_{FDR}$=0.01, $BF_{10}$=3.46) and education *("How much did your (familial) education influence your decision?")* (Agency: r=0.49, $p_{FDR}$=0.005, $BF_{10}$=22.56; Empathy: r=0.51, $p_{FDR}$=0.003, $BF_{10}$=33.68).

For each significant correlation between the %Pro_disob and an emotional measure or disobedience's criterion, we performed a segmented regression model analysis to investigate whether the relationship differed between military officer cadets and civilians. The analysis considered both directions of the relationship between the variables (i.e., the influence of %Pro_disob on the subjective measure: Subjective measure~%Pro_disob x Population; and the influence of the subjective measure on the %Pro_disob: %Pro_disob~Subjective measure x Population). Feelings of guilt and sadness, along with the perception of shock's painfulness, showed a significant influence of the Population in both runs. The results indicated that %Pro_disob and the three emotional measures influenced each other, with the effect of Population being significant for both directions (statistics are reported in Table 1). These findings showed stronger relationships in military officer cadets than civilians (Fig 4). However, no effect of Population was found for the feeling of responsibility, suggesting a similar relationship with the %Pro_disob in military officer cadets and civilians (Table 1, Fig 4). Regarding the three disobedience criteria (morality, sensitivity, education), only the sensitivity criterion showed a significant relationship with %Pro_disob influenced by the Population (Table 1, Fig 5), consistent with the results observed for the subjective feelings. The criteria of morality and education appeared to be positively associated with the rate of disobedience in a similar manner between civilians and military officer cadets.

**Table 1. Statistics (t and p values) for the segmented regression model analysis.**

**Model: %Pro_disob~Subjective measure x Population**

| Subjective measure: | Agency run | Empathy run |
| --- | --- | --- |
| **Feeling of responsibility** | t=0.79, p=0.43 | t=0.25, p=0.81 |
| **Feeling of guilt** | -- | t=3.76, p<0.001 |
| **Feeling of sadness** | t=4.07, p<0.001 | t=4.50, p<0.001 |
| **Perception of shock's painfulness** | t=1.93, p=0.06 | t=1.97, p=0.05 |
| **Morality criterion** | t=1.03, p=0.31 | t=0.13, p=0.90 |
| **Sensitivity criterion** | t=2.19, p=0.03 | t=1.71, p=0.09 |
| **Education criterion** | t=0.82, p=41 | t=0.80, p=0.43 |

**Model: Subjective measure~%Pro_disob x Population**

| Subjective measure: | Agency run | Empathy run |
| --- | --- | --- |
| **Feeling of responsibility** | t=0.87, p=0.39 | t=0.13, p=0.90 |
| **Feeling of guilt** | t=2.37, p=0.02 | t=2.61, p=0.01 |
| **Feeling of sadness** | -- | t=3.58, p<0.001 |
| **Perception of shock's painfulness** | t=1.58, p=0.12 | t=1.54, p=0.12 |
| **Morality criterion** | t=0.57, p=0.57 | t=−0.18, p=0.86 |
| **Sensitivity criterion** | t=1.98, p=0.05 | -- |
| **Education criterion** | t=0.58, p=0.56 | t=0.63, p=0.53 |

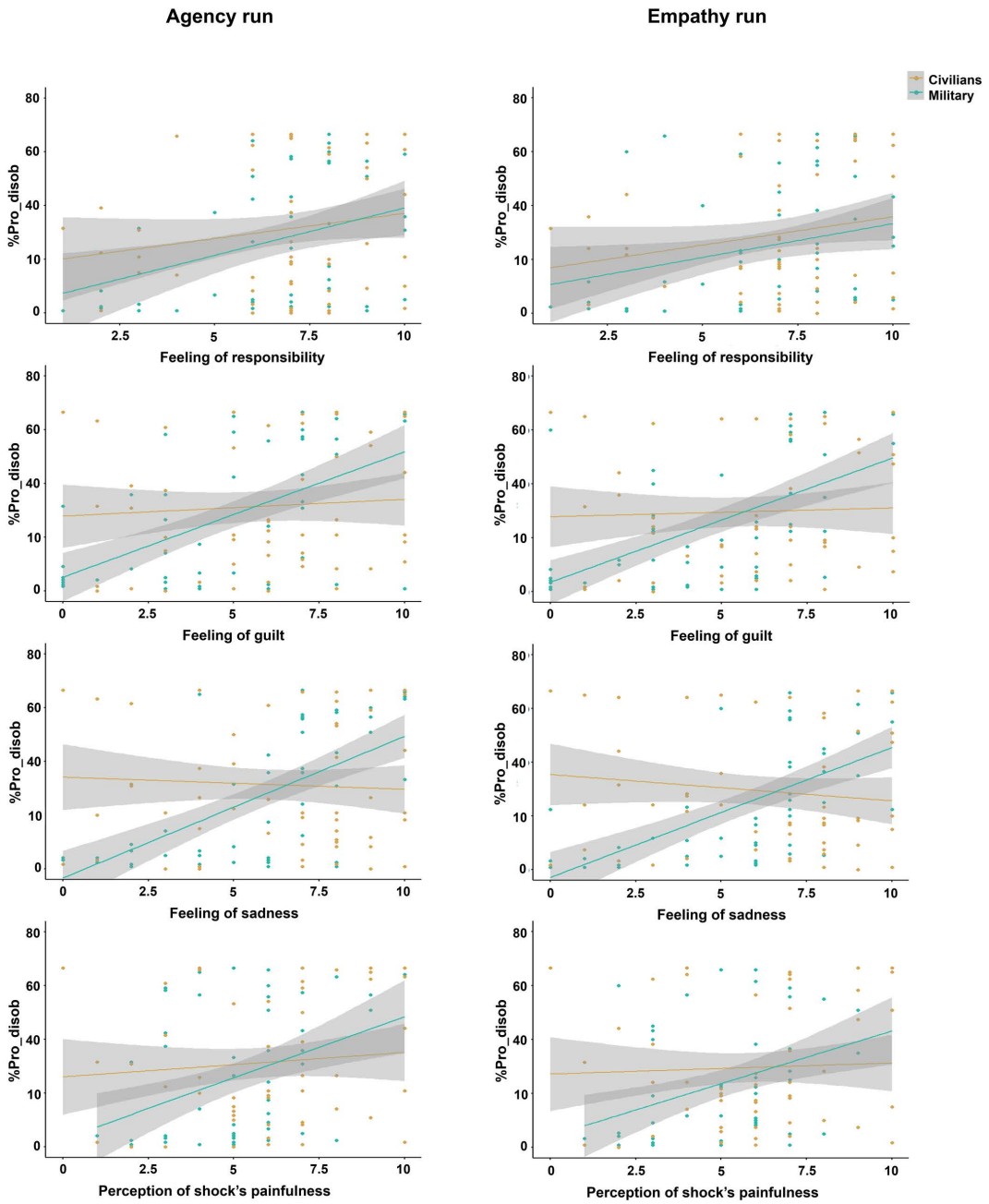

**Fig 4. Segmented regression model analysis between the subjective feelings (feeling of responsibility, feeling of guilt, feeling of sadness, perception of shock's painfulness) and %Pro_disob, taking the Population factor (Civilians, Military) into account.** Except for the feeling of responsibility, a Population effect was found for the majority of the investigated models, suggesting that the positive relationship between %Pro_disob and the subjective feelings was stronger in military than civilians. These plots only concern the %Pro_disob~Subjective measure x Population model. The corresponding plots of the Subjective measure~%Pro_disob x Population model are provided in S3 File.

The segmented regression model analysis investigated the Population effect on the relationships between the %Pro_disob and the subjective measures (emotional measures and disobedience's criteria) for each run. Only subjective measures that showed a significant correlation with the %Pro_disob were included in the analysis. Regression analyses were

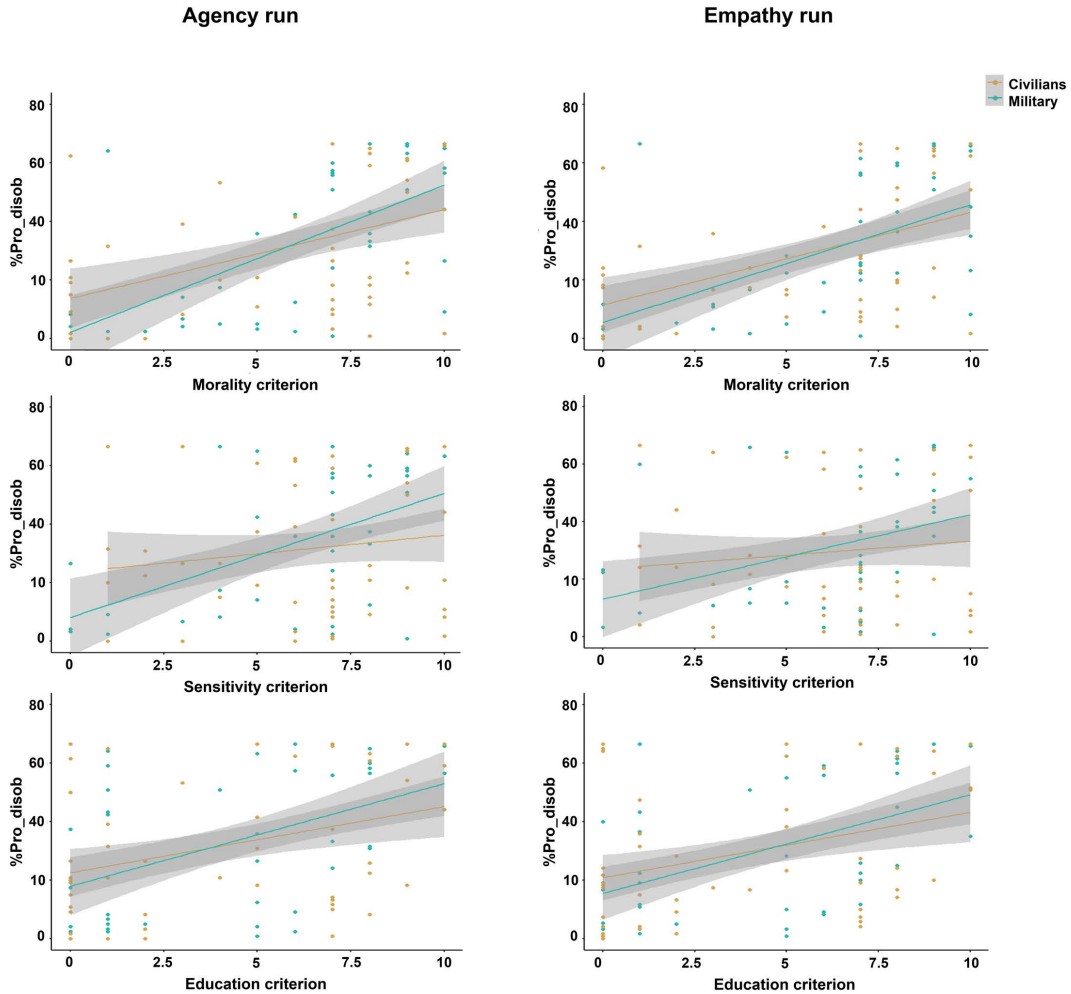

**Fig 5. Segmented regression model analysis between the disobedience's criteria (morality, sensitivity, education) and %Pro_disob, taking the Population factor (Civilians, Military) into account.** A Population effect was only found for sensitivity criterion suggesting that the positive relationship between %Pro_disob and the sensitivity criterion was stronger in military officer cadets than civilians. Morality and education criteria appeared to be positively associated with %Pro_disob in a similar way between civilians and military officer cadets. These plots only concern the %Pro_disob~Subjective measure x Population model. The corresponding plots of the Subjective measure~%Pro_disob x Population model are provided in S3 File.

conducted for each model, investigating the two possible directions of the relationship between the variables. -- indicates that the model could not be properly specified due to the independent variable containing too many duplicate values.

### fMRI results

**fMRI results at whole-brain level.** The four different contrasts of interests (and the associated reverse contrasts) were defined to target the epochs of our neuro-cognitive processes of interest during different phases of decision-making (i.e., pre-decision, decision, and post-decision). First, we used a global contrast ["Send a shock" > "Do not send a shock"] to investigate the processing of the auditory instructions. For the second contrast (i.e., ["Send a shock"/Obedience > "Send a shock"/Disobedience]), we specifically focused on trials involving immoral orders, that is, where the experimenter instructed agents to send a shock, and we contrasted trials in which agents obeyed that order (i.e., antisocial obedience trials) or disobeyed that order (i.e., prosocial disobedience trials). In a third contrast, (i.e., ["Send a

shock"/Obedience > "Do not send a shock"/Obedience]), we focused on obedience trials only and contrasted the received order. Finally, using the contrast (i.e., ["Send a shock"/Obedience + "Do not send a shock"/Disobedience]> ["Do not send a shock"/Obedience + "Send a shock"/Disobedience]), we focused on the brain processing when victims experienced pain or no pain. By conducting analyses on military officer cadets only, we largely replicated the results described for civilians in Tricoche, Rovai and Caspar, 2024 [2]. These results are detailed in S4 File. Overall, they showed that the act of sending a shock compared to not doing it (whatever the decision type, i.e., obedience or disobedience) involved the IOG/AG during the pre-decision-making phase, extending to the TPJ and SMG during the decision-making, and encompassing multiple fronto-parietal regions during the post-decision phase, including median ones.

The subsequent analyses focus on the comparison between the two populations.

The two-sample analysis revealed a marginal cluster for the ["Send a shock"/Disobedience > "Send a shock"/Obedience] contrast, but only for the post-decision phase targeting the outcome. Higher activity was found in the left caudate during disobedience compared to obedience when sending a shock in civilians compared to military officer cadets ([−16 21 3], Z = 3.99, cluster size = 84, p = 0.08). By lowering the statistical voxel-wise probability threshold to p = 0.005 uncorrected, the left caudate reached significance ([−16 21 3], Z = 3.99, cluster size = 264, p = 0.02) and the network extended on the left hemisphere to the left Superior Parietal Lobule (SPL; [−27–65 36], Z = 4.04, cluster size = 459, p = 0.001) and the left Inferior Occipital Gyrus (IOG; [−27 −94 −1], Z = 3.97, cluster size = 839, p = 0.03). However, as we contrasted trials in which the victim received a shock with those in which no shock was delivered, these results for the post-decision phase are considered unreliable and therefore difficult to interpret. The absence of significant cluster for the other decision-making phases suggests a comparable network between the two populations. A conjunction analysis supported this statement (Fig 6).

No other contrast (i.e., ["Send a shock"/Obedience > "Do not send a shock"/Obedience], ["Send a shock"/Obedience + "Do not send a shock"/Disobedience]> ["Do not send a shock"/Obedience + "Send a shock"/Disobedience], ["Send a shock" > "Do not send a shock"]), regardless of the phase of decision-making, showed significant clusters. The associated conjunction maps are presented in S5 File, which indicate, in line with these analyses, a comparable network between the two populations.

### Correlations between ROIs and the decision

**Military officer cadets.** For each epoch (pre-decision, decision-making, post-decision outcome and post-effects) we extracted the mean beta value of each of the 17 defined ROIs using the associated ["Send a shock"/Obedience > "Send a shock"/Disobedience] contrast and the ["Send a shock"/Obedience > "Do not send a shock"/Obedience] contrast. The contrast ["Send a shock"/Obedience > "Send a shock"/Disobedience] was used to investigate if a change of activity during the pre- and decision-making phases could result in different choices, leading to more or less prosocial disobedience rates. The contrast ["Send a shock"/Obedience > "Do not send a shock"/Obedience] was used to investigate if military officer cadets who maintained brain activity during the post-decision phase after sending a shock, could be more susceptible to disobey. We then conducted multiple correlation analyses between the beta values and the %Pro_disob of military officer cadets.

For the ["Send a shock"/Obedience > "Send a shock"/Disobedience] contrast, no correlation was found between the %Pro_disob and the ROI activity during the pre-decision phase. During the decision-making phase, four ROIs were positively significantly correlated with %Pro_disob: bilateral AI (left: r = 0.49, $p_{FDR}$=0.05, $BF_{10}$=0.48; right: r = 0.52, $p_{FDR}$=0.05, $BF_{10}$=24.46), right TPJ (r = 0.48, $p_{FDR}$=0.05, $BF_{10}$=5.97) and Prec/PCC (r = 0.46, $p_{FDR}$=0.05, $BF_{10}$=2.43). A positive trend for correlation with the left SMG was also found (r = 0.42, $p_{FDR}$=0.07, $BF_{10}$=3.86). Particularly, Bayes factors supported H1 for the right AI, right TPJ and left SMG. These correlations indicated that the more military officer cadets had activity in the AI, right TPJ, Prec/PCC and left SMG when making their decision to obey the order to send a shock, the more frequently they refused the orders to send a shock. This result could also be interpreted in another way: military officer cadets who disobeyed less had reduced activity in regions associated to the decision-making phase.

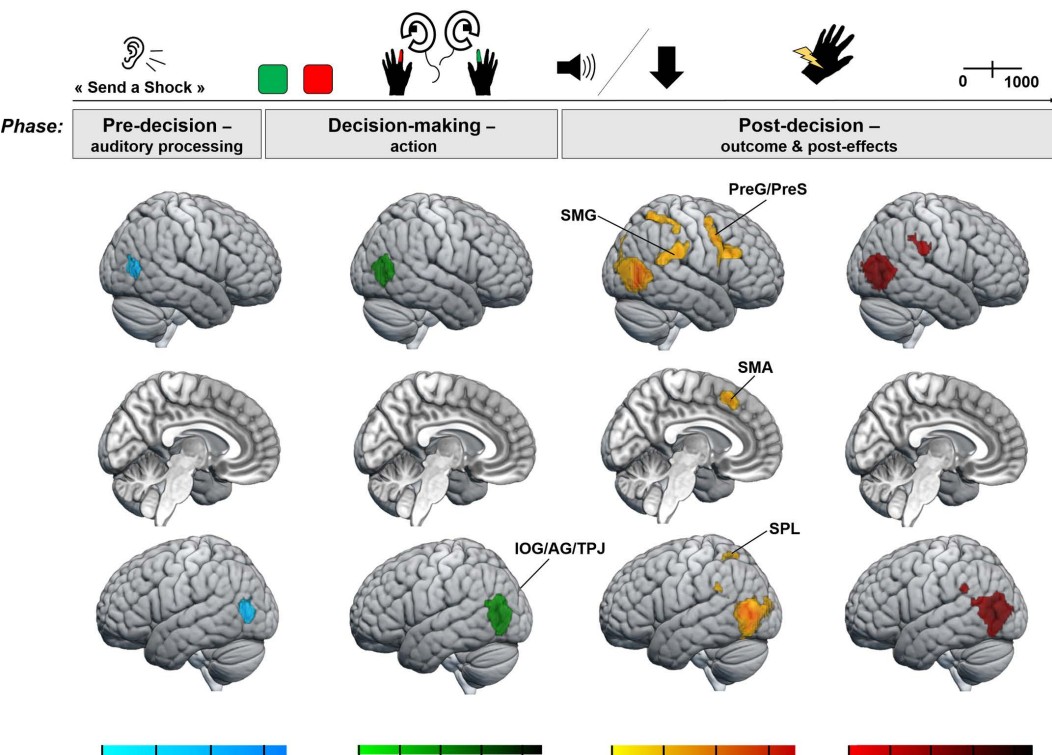

**Conjunction Military & Civilians: ["Send a shock"/Obedience > "Send a shock"/Disobedience]**

**Fig 6. Conjunction analysis between the two populations (Civilians, Military) for the ["Send a shock"/Obedience > "Send a shock"/Disobedience] contrast across all decision-making phases.** These brain maps indicate that both groups engaged a similar network, particularly the IOG/AG extending to TPJ, SMG and SPL, throughout the trial to obey rather than disobey orders.

For the ["Send a shock"/Obedience > "Do not send a shock"/Obedience] contrast targeting obedience trials only, we found positive correlations between the %Pro_disob and ROI activity during the post-decision phase. Specifically, for the post-decision outcome period, five ROIs, preferentially in the right hemisphere, were found to positively correlate with %Pro-disob: right IOG/AG ($r = 0.49$, $p_{FDR} = 0.01$, $BF_{10} = 13.53$), right SMA ($r = 0.45$, $p_{FDR} = 0.02$, $BF_{10} = 11.16$), right TPJ ($r = 0.41$, $p_{FDR} = 0.03$, $BF_{10} = 4.93$), right PreG ($r = 0.37$, $p_{FDR} = 0.04$, $BF_{10} = 13.28$) and Prec/PCC ($r = 0.43$, $p_{FDR} = 0.02$, $BF_{10} = 5.01$). All Bayes factors were in favor of H1. Finally, for the post-effects period, we again found a significant correlation and a Bayes factor supporting H1, with right IOG/AG ($r = 0.48$, $p_{FDR} = 0.01$, $BF_{10} = 5.23$), right PreG ($r = 0.44$, $p_{FDR} = 0.02$, $BF_{10} = 31.24$), right TPJ ($r = 0.42$, $p_{FDR} = 0.02$, $BF_{10} = 19.91$) and Prec/PCC ($r = 0.48$, $p_{FDR} = 0.01$, $BF_{10} = 67.68$). The left TPJ was also significant ($r = 0.43$, $p_{FDR} = 0.02$, $BF_{10} = 8.13$) and a trend was found for the right AI ($r = 0.36$, $p_{FDR} = 0.06$, $BF_{10} = 15.05$). These correlations indicated that the more military officer cadets had activity in these regions during this post-decision phase when they obeyed orders to send a shock, the more frequently they refused the orders to send a shock. This result could also be interpreted in another way: military officer cadets who disobeyed less had reduced activity in regions associated to the post-decision phase.

**Comparison between military officer cadets and civilians.** For each significant correlation between the %Pro_disob and ROI activity, we conducted a segmented regression model analysis to investigate whether the above-mentioned relationships were different between military officer cadets and civilians. The analysis was conducted considering the two directions of the relationship between the variables (i.e., the influence of %Pro_disob on the ROI activity: ROI

activity ~ %Pro_disob x Population; and the influence of the ROI activity on the %Pro_disob: %Pro_disob ~ ROI activity x Population).

For the decision-making phase investigated using the ["Send a shock"/Obedience > "Send a shock"/Disobedience] contrast, we found a Population effect on the two directions of the relationship between the %Pro_disob and the right TPJ as well as Prec/PCC's activity (ROI activity ~ %Pro_disob x Population: p = 0.04 for right TPJ and p = 0.01 for Prec/PCC; %Pro_disob ~ ROI activity x Population: p = 0.03 for right TPJ and p = 0.02 for Prec/PCC). We also found an influence on the relationship between the %Pro_disob and bilateral AI activity, but only considering the ROI activity ~ %Pro_disob x Population model (left AI: p = 0.08; right AI: p = 0.02). All these results indicated that the positive relationships between the %Pro_disob and the activity in right TPJ, Prec/PCC and bilateral AI, were stronger in military officer cadets compared to civilians (Fig 7). As an exploratory analysis, we conducted segmented regressions between the activity of these ROIs and disobedience-related criteria. We primarily focused on the three criteria that were significantly correlated with %Pro_disob (sensitivity, morality, and education). Sensitivity and education criteria did not show significant relationships with brain activity in any ROI, nor did they show interactions with the population factor. However, a stronger negative relationship was observed between the morality criterion and activity in the left AI in military officer cadets compared to civilians. This suggests that the increased recruitment of the left AI during disobedience among military officer cadets, relative to civilians, was not primarily explained by higher self-reported morality criterion. Full results for the other disobedience-related criteria are provided in S6 File. Interestingly, these analyses revealed stronger associations between right TPJ and Prec/PCC activity and both the "country's history" and "rejection of authority" criteria in military officer cadets compared to civilians. For the rejection of authority criterion, a significant relationship was also observed with left AI activity. Taken together, these findings suggest that in military officer cadets, the greater recruitment of these brain regions during decision-making, associated to higher prosocial disobedience, is also linked to higher subjective importance attributed to rejecting authority and acknowledging the historical role of their country.

For the post-decision phase targeting outcomes and post-effects, investigated using the ["Send a shock"/Obedience > "Do not send a shock"/Obedience] contrast, we did not find significant influence of the Population on the relationship between the ROI activities and the %Pro_disob. It suggests that the positive relationship between the %Pro_disob and the brain network associated with the post-decision phase when obeying orders, was consistent across both populations (Fig 7).

Results obtained for ROIs which did not show significant correlation with %Pro_disob in military officer cadets, are detailed in S7 File. Specifically, they indicated a stronger positive relationship between %Pro_disob and activity in the dmPFC for civilians compared to military officer cadets during the pre-decision phase. This stronger relationship in civilians was also observed with left TPJ during the decision-making phase, and with the left SMG, dmPFC, vmPFC/ACC and left SPL during the post-decision phase targeting post-effects.

## Discussion

In the present study, we investigated the neuro-cognitive processes underlying prosocial disobedience—refusing to deliver a shock to a third party—by focusing specifically on a military population accustomed to environments where authority plays a significant role, often daily, and where hierarchy is crucial. By comparing these military officer cadets with civilian participants from a previously collected dataset [2], we aimed to highlight the impact of this unique environment on neuro-cognitive processes involved across the decision-making phases, from pre-decision (when individuals receive the order) to post-decision (when they observe and integrate the consequences of their decision and action).

### Behavioral results: Disobedience among military officer cadets

Although the orders in this study did not come from a military superior, the experimenter's role as an authority figure within the experimental setting likely created a similar context for the obedience/disobedience process that military officer cadets

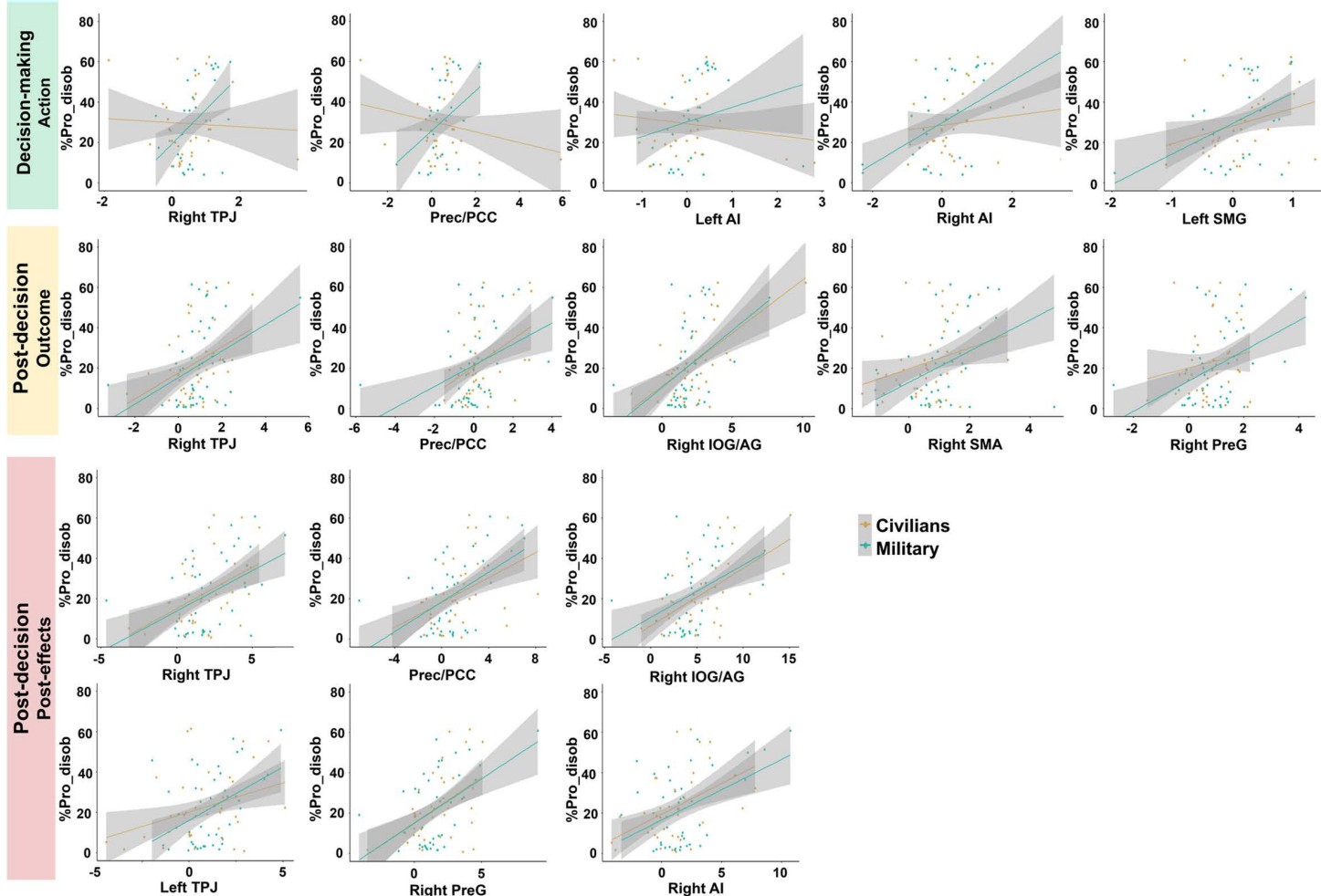

**Fig 7. Segmented regression model analysis between the ROIs activity and %Pro_disob, taking the Population factor (Civilians, Military) into account.** This analysis was conducted only when a significant correlation was found between the %Pro_disob and a ROI (e.g., right AI). A Population effect was found only during the decision-making phase. It suggests that the positive relationship between %Pro_disob and the right TPJ, Prec/PCC, and bilateral AI, was stronger in military officer cadets than in civilians. However, no population effect was found for the post-decision phase suggesting similar relationships. These plots pertain only to the %Pro_disob ~ ROI activity x Population model. Corresponding plots of the ROI activity ~ %Pro_disob x Population model are provided in S2 Fig.

regularly face. Within this framework, the results reveal that military officer cadets, like civilians, are indeed capable of defying authority, ultimately disobeying orders. All military officer cadets engaged in prosocial behavior at various points by refusing, to varying extents, to deliver a shock to the victim's hand. However, military officer cadets exhibited lower levels of disobedience compared to civilians [2]. Their responses were also more homogeneous, with fewer ultra-altruistic profiles (i.e., high levels of prosocial disobedience) and the absence of antisocial profiles, as no military officer cadet repeatedly delivered shocks for additional money when the experimenter did not request it. The motivations for disobedience among military officer cadets appeared similar to those reported by civilians. Participants primarily refused to deliver shocks for moral reasons, as well as due to their personal education. However, unlike civilians, military officer cadets' disobedience was also positively associated with sensitivity, suggesting an additional emotional component. This is further underscored by positive correlations between the percentage of prosocial disobedience and feelings of responsibility,

sadness, and guilt. Notably, the link with sadness and guilt was stronger in military officer cadets compared to civilians. Lastly, military officer cadets demonstrated a positive association between their disobedience rate and the perceived pain of the shocks, as reported both during debriefing and on the pain scale rated throughout the task. This may suggest that military officer cadets who chose to disobey orders most frequently were those who associated negative emotions with delivering a shock. The absence of this effect among civilians implies that moral or responsibility-based criteria—likely shaped by their upbringing—are sufficient to lead to disobedience. In contrast, military officer cadets appeared to require an additional, more personal emotional component to make the decision to disobey. Emotions, particularly empathic ones, may have served as a "beneficial guide," motivating individuals to make decisions when confronted with situations of injustice [35].

### fMRI results: A replication of the engaged processes during the task

To compare military officer cadets and civilians, we used the exact same paradigm developed in Tricoche et al., 2024, and the same MRI scanner. The results obtained in the present study allowed us to replicate the findings previously described in civilians, thereby strengthening their interpretations. Whole-brain results indicated a similar brain network between the two populations. During the pre-decision phase, the IOG/AG was more active when participants were instructed to send a shock compared to not sending a shock. This activity extended bilaterally across the TPJ and SMG during the decision-making phase and until the action of sending or not sending the shock. In the post-decision phase, when agents observed the consequences of their action, a fronto-parietal network—encompassing additional areas such as the SPL, PreG/PreS, IFG, as well as the Prec and SMA in the median region—was more activated if the victim received a shock. These findings suggest identical processes between the two groups, though the level of engagement of these processes may vary between them (see below). Specifically, both the IOG/AG and TPJ, key regions in moral judgments, were actively engaged throughout the trial, from the receipt of the auditory order to the witnessed outcome [17,18]. Meanwhile, the post-decision phase particularly involved social brain regions associated with empathy for pain and the feeling of guilt [7].

### fMRI results: Distinctions between military officer cadets and civilians before deciding to disobey

Focusing on the military officer cadets group, ROI analyses during the decision-making phase showed a positive correlation between the prosocial disobedience rate and activity in bilateral AI, right TPJ, Prec/PCC, and left SMG when obeying an order to send a shock. Except for the left SMG, these relationships were stronger in military officer cadets compared to civilians, suggesting a similar network but with different levels of engagement. It appeared that military officer cadets required greater recruitment of these regions, particularly the right TPJ and Prec/PCC, associated with mentalizing [36–38], and the AI associated with emotions [39], including social-related emotions [40]. The AI has also been implicated in self-reflection processing as opposed to other-reflection [41], a function indirectly supported by Prec/PCC's involvement in self-awareness [42]. In our case, this might indicate the military officer cadets' ability to prioritize their own judgment over the experimenter's perspective in order to resist the received order. Additionally, the right TPJ has also been linked to reorienting of attention as well as explicit SoA during social tasks, supporting an interpretation related to responsibility or personal awareness [43–45]. Interestingly, we also found a positive relationship between the activity of these brain regions and the subjective importance attributed to rejection of authority and the historical role of the country. This suggests that, for military officer cadets, the successful enactment of prosocial disobedience may be supported not only by cognitive-emotional mechanisms, but also by broader socio-cultural values that reinforce critical evaluation of authority. Moreover, our behavioral results indicated a lower SoA when military officer cadets disobeyed, compared to when they obeyed—suggesting a reduced sense of authorship over their actions when defying instructions from an authority figure, as opposed to following orders. This may be because, for military personnel, resisting orders is not a conditioned behavior due to their training. However, correlational analyses showed that the more military officer cadets disobeyed, the

more their SoA increased. This result, combined with the observed neural activities, suggests that military officer cadets may had a lower natural tendency to disobey; to do so, they seemed to engage emotional regions and focused on self-reflection, thereby enhancing their sense of responsibility and agency, which ultimately enabled disobedience. Finally, disrupting the right TPJ using transcranial stimulations (TMS or tDCS) were found to reduce mental state ability during moral judgment, when the scenario induced harming someone else [1,46], revealing its crucial role in prosocial moral behaviors.

Civilians, on the other hand, showed increased dmPFC activity related to disobedience in the pre-decision phase and greater left TPJ activity during decision-making, compared to military officer cadets. Civilians therefore engaged different mentalizing network regions than military officer cadets. Rilling and Sanfey suggest that the prefrontal cortex plays a crucial role in social decision-making [47]. Specifically, mPFC activation has been observed in situations where others might hold thoughts about the individual, for example, if they are evaluating them [48–50]. Individuals may form impressions of how others perceive them, that is a self-description based on imagined perceptions from others. Similarly, the left TPJ has a role in taking account the gap between self and other perspectives [51]. A possible interpretation, then, is that civilians were more sensitive than military agents to how others perceived them, especially concerning how the victim might view them, leading to a higher rate of disobedience.

Overall, by distinguishing the regions preferentially activated in military officer cadets versus civilians—specifically with the right TPJ and AI more engaged in the former, and the left TPJ and dmPFC in the latter—we suggest a slightly different form of mentalization between the two populations. When disobeying, civilians may employ a more cognitive and analytical form of mentalization, distinguishing between their own and others' perspectives (i.e., cognitive empathy). In contrast, military officer cadets may engage a more emotional form (i.e., affective empathy), attempting to put themselves in others' emotional shoes and feeling accountable for their actions when they negatively impact others [52]. Of course, these interpretations, both for civilians and military officer cadets, should be approached cautiously to avoid reverse inference, particularly because these regions are involved in numerous functions.

### fMRI results: Distinctions between military officer cadets and civilians in disobeying when witnessing a shock

During the post-decision phase, when agents witnessed the consequences of their choice—specifically when the victim received a shock—correlation differences in brain activity associated with overall disobedience rates were observed between civilians and military officer cadets. Interestingly, these differences were lateralized. For military officer cadets, positive correlations were primarily shown with regions in the right hemisphere, including the IOG/AG, TPJ, SMA, PreG, and Prec/PCC. Disobedience rates were also correlated with increased activity in the left TPJ. However, comparison with civilians showed no significant difference in the strength of these relationships. By contrast, civilians exhibited stronger positive correlations than military officer cadets in several left-hemisphere regions, including the TPJ, SMG, and SPL. Additionally, medial regions—dmPFC and vmPFC/ACC—showed stronger positive correlations with disobedience rates in civilians than in military officer cadets. Thus, similar to the decision-making phase, during the post-decision phase, military officer cadets appeared to rely more on the right hemisphere, while civilians activated both hemispheres and prefrontal medial regions, facilitating prosocial disobedience. These findings support the hypothesis of cognitive empathy in civilians and affective empathy in military officer cadets, which future studies should investigate in greater detail.

### Limitations and future studies

One distinction between civilians and military officer cadets is that while they both received a financial reward for each shock sent to the victim, civilians earned the money for themselves, and military officer cadets earned it for a charity project. This may have influenced their decisions during the task.

In an ideal scenario, we would have preferred not mentioning at all the possibility to disobey to our participants, even minimally, to be closer to a decision internally generated. However, as pilot studies showed that participants never

disobeyed in this fMRI setup, mentioning this did not want to alter our study; we had no other option to obtain a reliable prosocial disobedience rate for analyses.

A previous study conducted on how the SoA differs between different military officer cadets and civilians showed that differences compared to civilians are stronger for military personnel with lower decisional power, such as privates or junior cadets, compared to officers. However, given the time required for participants to complete the task (i.e., leaving the military institution to cross the city to go to the hospital where the MRI scanner was located, and the duration of the task), we did not receive authorization to involve military personnel of different ranks. Future studies could consider involving military personnel with different levels of decisional power to evaluate if they also rely on similar brain processes to disobey orders.

## Conclusion and future directions

Overall, our results suggest that one's environment can influence moral decisions and responses to orders from an authority figure. While we observed that the engaged processes are consistent across civilians and military officer cadets, the degree of engagement varied, with observed neural differences indicating distinct "strategies" for disobedience between military officer cadets and civilian participants. Notably, we observed that military officer cadets had a lower sense of agency when they disobeyed orders, even when they did so prosocially, compared to civilians. While previous studies tend to suggest that positive outcomes enhance the sense of agency [53,54], the reduction of SoA in military officer cadets for prosocial actions may suggest that disobeying is a more complex decision for them, for which they do not take full authorship, at least as measured with implicit measures. We also observed that military officer cadets relied more than civilians on emotional brain regions associated with the decision and post-decision phases, perhaps as a compensatory mechanism for the lower SoA. Additional studies are necessary to better understand how the processes involved in the different decision phases interact with each other and how they can compensate for each other's involvement to achieve disobedience. These findings have practical implications, particularly in military and institutional contexts, including training and ethics education. While military personnel are taught that they are legally required to refuse unlawful orders under international humanitarian law (United Nations, 1950), little attention is typically given to the psychological and neurocognitive mechanisms that make such disobedience difficult to enact. The present study suggests that prosocial disobedience is not only a legal or moral obligation, but also a cognitively and emotionally demanding act, particularly for individuals embedded in hierarchical structures. Integrating our insights into training programs could be valuable: for example, by combining legal instruction with practical exercises that enhance soldiers' awareness of the psychological and neural processes involved in disobedience, including how obedience can become a conditioned response, and how emotional and cognitive strategies may be recruited to overcome it. By highlighting these mechanisms, training could help normalize disobedience in situations where moral and law require it—not as defiance, but as a deliberate and reasoned act of ethical responsibility. Future work could explore how such training might increase the likelihood of prosocial disobedience and support a more balanced sense of agency among military personnel when they face ethically challenging decisions.

## Supporting information

**S1 File. Methodology and results of the linear regression analyses.** These analyses were conducted to investigate which individual characteristics, assessed by questionnaires, could best explain prosocial disobedience.
(DOCX)

**S2 File. Individual response profiles for each agent and each run (Agency, Empathy) & percentage disobedience analysis.** Response profiles expressed as percentages (%) for each of the four conditions ("Send a shock"/Obedience, "Send a shock"/Disobedience, "Do not send a shock"/Obedience, "Do not send a shock"/Disobedience). A supplementary Instruction x Choice x Run x Population ANOVA on the percentage choice was conducted. For this analysis, the

percentage was calculated based on the number of trials per instruction ("Send a shock" = 80 trials, "Do not send a shock" = 40 trials).
(DOCX)

**S1 Fig. Cumulative %Pro_disob.** Individual plots (left side) and group plots (right side) representing the cumulative %Pro_disob (up to 80, as the instruction to send a shock was given 80 times) over the 120 trials for both Agency (top) and Empathy (bottom) runs. Participants were generally consistent across the two runs in their decision-making strategies.
(TIF)

**S2 Fig. Plots of the segmented regression model analyses with ROIs.** Segmented regression model analysis between the ROIs and %Pro_disob, taking the Population factor (Civilians, Military) into account. These plots concern the ROI's activity ~ %Pro_disob x Population model. An effect of Population was only found for the decision-making phase, suggesting that the positive relationship between %Pro_disob and the right TPJ, Prec/PCC, and bilateral AI, was stronger in military participants than civilians.
(TIF)

**S3 File. Segmented regression model analyses with questionnaires.** A: Segmented regression model analysis between the subjective feelings (feeling of responsibility, feeling of guilt, feeling of sadness, perception of shock's painfulness) and %Pro_disob, taking the Population factor (Civilians, Military) into account. Except for the feeling of responsibility, a Population effect was found for the majority of the investigated models suggesting that the positive relationship between %Pro_disob and the subjective feelings, was stronger in military participants than civilians. These plots concern the Subjective measure ~ %Pro_disob x Population model. B: Segmented regression model analysis between the disobedience criteria (morality, sensitivity, education) and %Pro_disob, taking the Population factor (Civilians, Military) into account. A Population effect was only found for sensitivity criterion suggesting that the positive relationship between %Pro_disob and the sensitivity criterion was stronger in military than civilians. Morality and education criteria appeared to be positively associated with %Pro_disob in a similar way between civilians and military participants. These plots concern the Subjective measure ~ %Pro_disob x Population model.
(DOCX)

**S4 File. fMRI results at the whole-brain level for military participants.** We used four contrasts of interest: 1) the ["Send a shock" > "Do not send a shock"] instruction contrast, 2) the ["Send a shock"/Disobedience > "Send a shock"/Obedience] decision type contrast, 3) the ["Send a shock"/Obedience > "Do not send a shock"/Obedience] obedience contrast and 4) the ["Send a shock"/Obedience + "Do not send a shock"/Disobedience] > ["Do not send a shock"/Obedience + "Send a shock"/Disobedience] outcome contrast. For the second contrast we also applied an exclusive mask using the significant clusters obtained from the ["Send a shock"/Obedience > "Do not send a shock"/Obedience] contrast. This mask filtered out all significant voxels identified in the ["Send a shock"/Obedience > "Do not send a shock"/Obedience] contrast, to retain only voxels representing brain activity associated with disobedience in the ["Send a shock"/Disobedience > "Send a shock"/Obedience] contrast. The significance threshold was set at p < 0.05 (FWE corrected for multiple comparisons) at the cluster level, with an initial voxel-wise probability threshold of p < 0.001 uncorrected; except for the analysis using the exclusive mask which was reported with a threshold of p < 0.005 uncorrected.
(DOCX)

**S5 File. Conjunction maps.** Conjunction maps between military participants and civilians for the three other contrasts of interest: ["Send a shock"/Obedience > "Do not send a shock"/Obedience] (Fig A), ["Send a shock" > "Do not send a shock"] (Fig B) and ["Send a shock"/Obedience + "Do not send a shock"/Disobedience] > ["Do not send a shock"/Obedience + "Send a shock"/Disobedience] (Fig C). The first contrast focusing on obedience trials by contrasting the order received, was investigated for all time windows. The reverse contrast did not reveal any clusters. The contrast ["Send a

shock" > "Do not send a shock"] and its reverse contrast, investigating the processing of the auditory instructions, were only applied during the pre-decision phase. Finally, the contrast comparing trials where the victim received pain or no pain, was studied between the decision-making and the post-decision periods. The reverse contrast did not reveal any clusters. (DOCX)

**S6 File. Segmented regression model analyses between disobedience's criteria, significant ROIs, and population.** Segmented regression model analysis between the ROIs (i.e., bilateral AI, right TPJ and Prec/PCC) and disobedience criteria (i.e., shock quantity, money, entertainment, rejection of authority, experimental, fear of judgment, and country's history criterions), taking the Population factor (Civilians, Military) into account. Both models are considered: ROI's activity ~ criterion x Population (model 1) and criterion ~ ROI's activity x Population (model 2). (DOCX)

**S7 File. Segmented regression model analyses with no significant ROIs.** Segmented regression model analysis between the ROIs and %Pro_disob, taking the Population factor (Civilians, Military) into account. Both models are considered: ROI's activity ~ %Pro_disob x Population (model 1) and %Pro_disob ~ ROI's activity x Population (model 2). Only ROIs that showed no significant correlation with %Pro_disob for the military participants are analyzed below (results on the remaining ROIs are described in the main manuscript). (DOCX)

## Acknowledgments

The project, as part of the PET-MR project at the Hôpital Universitaire de Bruxelles (Hôpital Erasme), is supported by the Association Vinçotte Nuclear (AVN, Brussels, Belgium).

## Author contributions

**Conceptualization:** Leslie Tricoche, Salvatore Lo Bue, Emilie A. Caspar.

**Data curation:** Leslie Tricoche, Antonin Rovai.

**Formal analysis:** Leslie Tricoche, Antonin Rovai.

**Funding acquisition:** Emilie A. Caspar.

**Investigation:** Leslie Tricoche, Antonin Rovai, Salvatore Lo Bue.

**Methodology:** Antonin Rovai, Emilie A. Caspar.

**Project administration:** Salvatore Lo Bue, Emilie A. Caspar.

**Resources:** Emilie A. Caspar.

**Supervision:** Emilie A. Caspar.

**Validation:** Emilie A. Caspar.

**Visualization:** Leslie Tricoche.

**Writing – original draft:** Leslie Tricoche.

**Writing – review & editing:** Antonin Rovai, Salvatore Lo Bue, Emilie A. Caspar.

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
