## [Decision Letter · Decision Letter 0]

PONE-D-25-03927Neuro-Cognitive Specificities in Prosocial Disobedience: A Comparative fMRI Study of Civilian and Military PopulationsPLOS ONE

Dear Dr. Tricoche,

Thank you for submitting your manuscript to PLOS ONE. After careful consideration, we feel that it has merit but does not fully meet PLOS ONE’s publication criteria as it currently stands. Therefore, we invite you to submit a revised version of the manuscript that addresses the points raised during the review process.

**The reviewer comments, though primarily focused on minor details, still require careful attention.**

**Please address each point raised by the reviewers in a thorough and professional manner. These types of comments are standard in the peer review process and addressing them diligently will undoubtedly strengthen your manuscript.**

**I encourage you to work through these comments methodically, as doing so will result in a more polished final product that effectively communicates your important findings to the field.**

We look forward to receiving your revised manuscript.

Kind regards,

Rei Akaishi

Academic Editor

PLOS ONE

**Journal Requirements:**

1. When submitting your revision, we need you to address these additional requirements. Please ensure that your manuscript meets PLOS ONE's style requirements, including those for file naming. The PLOS ONE style templates can be found at https://journals.plos.org/plosone/s/file?id=wjVg/PLOSOne_formatting_sample_main_body.pdf and https://journals.plos.org/plosone/s/file?id=ba62/PLOSOne_formatting_sample_title_authors_affiliations.pdf 2. Thank you for stating the following financial disclosure: E.C: ERC Starting Grant DISOBEY (grant number: 101075690) E.C: BOF Starting Grant from Ghent University (grant number: BOF/STA/202109/025)  Please state what role the funders took in the study.  If the funders had no role, please state: "The funders had no role in study design, data collection and analysis, decision to publish, or preparation of the manuscript." If this statement is not correct you must amend it as needed. Please include this amended Role of Funder statement in your cover letter; we will change the online submission form on your behalf. 3. Thank you for stating the following in the Acknowledgments Section of your manuscript: The project was supported by an ERC Starting Grant DISOBEY (grant number: 101075690) and by a BOF Starting Grant from Ghent University (grant number: BOF/STA/202109/025) granted to Emilie A. Caspar. Views and opinions expressed are however those of the author(s) only and do not necessarily reflect those of the European Union or the European Research Council Executive Agency. Neither the European Union nor the granting authority can be held responsible for them. We note that you have provided funding information that is not currently declared in your Funding Statement. However, funding information should not appear in the Acknowledgments section or other areas of your manuscript. We will only publish funding information present in the Funding Statement section of the online submission form. Please remove any funding-related text from the manuscript and let us know how you would like to update your Funding Statement. Currently, your Funding Statement reads as follows: E.C: ERC Starting Grant DISOBEY (grant number: 101075690) E.C: BOF Starting Grant from Ghent University (grant number: BOF/STA/202109/025) Please include your amended statements within your cover letter; we will change the online submission form on your behalf. 4. Please amend either the abstract on the online submission form (via Edit Submission) or the abstract in the manuscript so that they are identical. 5. Please include captions for your Supporting Information files at the end of your manuscript, and update any in-text citations to match accordingly. Please see our Supporting Information guidelines for more information: http://journals.plos.org/plosone/s/supporting-information.

**Additional Editor Comments:**

The reviewer comments, though primarily focused on minor details, still require careful attention.

Please address each point raised by the reviewers in a thorough and professional manner. These types of comments are standard in the peer review process and addressing them diligently will undoubtedly strengthen your manuscript.

I encourage you to work through these comments methodically, as doing so will result in a more polished final product that effectively communicates your important findings to the field.

Reviewers' comments:

Reviewer's Responses to Questions

**Comments to the Author**

1. Is the manuscript technically sound, and do the data support the conclusions?

Reviewer #1: Partly

Reviewer #2: Partly

2. Has the statistical analysis been performed appropriately and rigorously? 

Reviewer #1: Yes

Reviewer #2: No

3. Have the authors made all data underlying the findings in their manuscript fully available?

Reviewer #1: Yes

Reviewer #2: Yes

4. Is the manuscript presented in an intelligible fashion and written in standard English?

Reviewer #1: Yes

Reviewer #2: Yes

5. Review Comments to the Author

**Reviewer #1:**  In their manuscript, “Neuro-cognitive specificities in prosocial disobedience, A comparative fMRI study of civilian and military populations”, Tricoche and colleagues compare different behavioral and neural correlates of (prosocial) disobedience. The find that civilians and military participants differ with respect to the stated reasons for disobedience as well as the neural regions most activated during the decision to disobey.

The research question of which aspects and potential determinants of (prosocial) disobedience may differentiate a civilian and a military population is interesting and the data thus of general interest to the community. I also very much welcome the data sharing and preregistration of the project, the hypotheses, and the analyses.

However, in its current form, the manuscript would benefit from a lot of (re-)focussing especially regarding the introduction and I think the authors should be more prudent when interpreting the fMRI results to avoid jumping to conclusions. That said, I think the points I raise can all be well addressed.

I hope my comments will help clarify, streamline, and improve the manuscript to ultimately move the manuscript towards publication.

Major:

1) The introduction is currently hard to follow and it would be very helpful to streamline the information provided towards the specific research questions. In detail, it may be helpful to focus on the 2-3 studies that most closely resemble the current study design and summarize findings from additional studies very briefly to support the fact that there is work out there showing a specific thing. E.g. study 32 seems to be especially relevant but is only mentioned quite late in the introduction. This will also be help to avoid the listing of too many brain regions that may or may not be more or less relevant to the actual research question.

2) The first page of the introduction needs more references. E.g. which studies have started investigating disobedience with fMRI, EEG, and tDCS (line 49), or which studies show that obedience may hold a different significance to university students vs. military personel (line 51), also line 59 (coercion effect), and 61 (sentence ending with “e.g. prosocial disobedience”). Please be mindful of supporting your claims by references directly where you make the claim.

3) The whole work would benefit from more theoretical and conceptual grounding. For example, what are the exact assumed mechanisms that are expected to differ between the two population? Why exactly would that be the expectations? What are the assumed differences between resisting to an order and disobedience as investigated here (cf. line 92)? This will also greatly help with the interpretation of the fMRI results, I think.

4) Throughout the manuscript certain brain regions are labelled as e.g. “guilt-related” (ACC, TPJ, and basalganglia, line 75) or neural results from a previous study appear to be used to motivate behavioral predictions for the current one (line 146-149). I would strongly encourage the authors to be more mindful of the actual connection between activation in certain brain areas and behavior as especially regions such as the ACC and TPJ clearly are implicated in a variety of social as well as non-social cognitive processes. I also was surprised to read in lines 112/113 that left-lateralization “in response to immoral situations has been observed for the TPJ[…]” and one page later (line 128) "cathodal stimulation of the right TPJ showed an increased tendency to harm" an avatar. This looks like a contradiction to me. I am not opposed to lateralization per se, but in the present paradigm, I am not convinced it holds the key to a valuable understanding of the results.

5) I was surprised to see that the main behavioral DV is a percentage of disobedience but does not seem to be relative to the relevant number of trials. This seems make the analysis less informative than it couled be imo (“As the number of instructions so give a shock was higher than the order to not give a shock, the main effect of instruction was expected”). You really want to avoid this. The percentage should be more interepretable relative to the number of trials in which the instructions was to shock vs. not too shock rather than relative to the whole 120 trials as it appears to be now. I was also wondering, why the authors did not opt for the analysis of trial-by-trial data using, e.g. logistic mixed-effects models. I think this would be very beneficial and also facilitate interpretation of the results.

6) A further boost of explanatory evidence could be gained from relating the stated reasons for disobeying to the brain activation or contrasts of interest. This way, the results could paint a more comprehensive picture of which factors do or don’t influence differential neural correlates of disobedience in civilian and military participants.

Minor:

1) Please specify the shape and extent of the imaging ROIs (line 383). Did you use spheres or atlas-based ROIs, or something completely different? Helpful to know here.

2) It would be helpful to include a clearer explanation for the SoA measure and what exactly it means. I appreciate that the authors always included the information about what high and low values mean, respectively but I think it will be helpful to expand on this. I also gathered from section 3.1.2 that the SoA was lower for disobedience than for obedience decisions. This seems quite counter-intuitive, right? Would be helpful to discuss this.

3) I typically don’t use segmented regression and was not familiar with the concept until reviewing this paper. Keeping in mind that other readers (I also asked other methodologically interested colleagues) may also not be familiar with it, it may be helpful to include a one sentence explainer what it is and how the interpretation of the results relate to other types of regression (e.g. in segmented regression, the population effect appears to be a main effect whereas in linear regression, the same effect would have been an interaction effect – if understood correctly).

4) I recommend having a native speaker proof-read the manuscript. I spotted a few errors that could be weeded out to further improve readability of a revised version.

**Reviewer #2:**  Dear Authors,

Thank you to the authors for submitting a manuscript for consideration for publication in PLOS One. Motivated to investigate how the neurocognitive processes underlie prosocial disobedience, the manuscript reports research using functional magnetic resonance imaging (fMRI) on civilians and military personnel.

Please refer to the attached "Comments to the Authors" file for detailed comments on the manuscript.

Thank you.

6. PLOS authors have the option to publish the peer review history of their article (what does this mean? ). If published, this will include your full peer review and any attached files.

**Do you want your identity to be public for this peer review?** For information about this choice, including consent withdrawal, please see our Privacy Policy .

Reviewer #1: **Yes: ** Anne Saulin

Reviewer #2: No

---

## [Author Response · Author response to Decision Letter 1]

29 Apr 2025

Please find our responses to the reviewers and editor comments in the "Reponse to reviewers" document attached to this submission.

---

## [Decision Letter · Decision Letter 1]

PONE-D-25-03927R1Neuro-Cognitive Specificities in Prosocial Disobedience: A Comparative fMRI Study of Civilian and Military PopulationsPLOS ONE

Dear Dr. Tricoche,

Thank you for submitting your manuscript to PLOS ONE. After careful consideration, we feel that it has merit but does not fully meet PLOS ONE’s publication criteria as it currently stands. Therefore, we invite you to submit a revised version of the manuscript that addresses the points raised during the review process.

Thank you for your careful revision of the manuscript. While one of the original reviewers has raised some remaining concerns, these are not major issues. Please address these points in your response.

We look forward to receiving your revised manuscript.

Kind regards,

Rei Akaishi

Academic Editor

PLOS ONE

Journal Requirements:

Additional Editor Comments (if provided):

Thank you for your careful revision of the manuscript. While one of the original reviewers has raised some remaining concerns, these are not major issues. Please address these points in your response.

Reviewers' comments:

Reviewer's Responses to Questions

**Comments to the Author**

1. If the authors have adequately addressed your comments raised in a previous round of review and you feel that this manuscript is now acceptable for publication, you may indicate that here to bypass the “Comments to the Author” section, enter your conflict of interest statement in the “Confidential to Editor” section, and submit your "Accept" recommendation.

Reviewer #1: (No Response)

2. Is the manuscript technically sound, and do the data support the conclusions?

Reviewer #1: Yes

3. Has the statistical analysis been performed appropriately and rigorously? 

Reviewer #1: Yes

4. Have the authors made all data underlying the findings in their manuscript fully available?

Reviewer #1: No

5. Is the manuscript presented in an intelligible fashion and written in standard English?

Reviewer #1: Yes

6. Review Comments to the Author

Reviewer #1: I thank the authors for addressing most of the points put forwarded. Especially, the introduction has benefited a lot from having been revised!

Two points remain which are hopefully easily addressed:

1) I was not able to locate data and code following the link provided in the manuscript. Under the Files tab in the OSF project, I could only see the manusciprt pdf file and the supplement pdf file. From what I gather, data (and possibly code) should be available as e.g., csv or txt file. In the case of imaging data, as .nii probably. Thus, it would be helpful to point the reader to where the data is at in the data availability link. If it was hidden somewhere in the project, I apologize, but would still recommend to make them easy to find.

2) Regarding my previous point on using the 80 trials of instructed shock in place of the 120 trials that cover all trials, I was surprised to see that the authors presented a correlation between those two options. Mathematically, any difference from a correlation unequal to 1 would be a results of rounding errors and is beside the point I tried to make.

The key point I did aim to make was that the way the ANOVA is set up does not reflect the way the dependent variable is chosen. That is, if you include the factor of instruction (to shock vs. no to shock), the dependent variable should be instruction -specific. I think it would thus be helpful to include this analysis for both samples - even if only in the supplement - to demonstrate that the effects hold. An additional benefit would be that you would actually be able to interpret an interaction effect of decision X instruction.

I do appreciate that it won't be necessary to rerun all analysis with this new %pro_disob.

7. PLOS authors have the option to publish the peer review history of their article (what does this mean? ). If published, this will include your full peer review and any attached files.

**Do you want your identity to be public for this peer review?** For information about this choice, including consent withdrawal, please see our Privacy Policy .

Reviewer #1: No

---

## [Author Response · Author response to Decision Letter 2]

27 Jun 2025

Responses to the reviewer's comment are given in the attached rebuttal letter.

---

## [Decision Letter · Decision Letter 2]

Neuro-Cognitive Specificities in Prosocial Disobedience: A Comparative fMRI Study of Civilian and Military Populations

PONE-D-25-03927R2

Dear Dr. Tricoche,

We’re pleased to inform you that your manuscript has been judged scientifically suitable for publication and will be formally accepted for publication once it meets all outstanding technical requirements.

Kind regards,

Rei Akaishi

Academic Editor

PLOS ONE

Additional Editor Comments (optional):

Please address the remaining minor comments from the reviewer:

Thank you to the authors for addressing the remaining points!

I was able to easily find the data. Is the R code provided as well? Maybe double-check whether this is necessary, too.

I recommend checking the manuscript for language errors.

Other than these two easily to address very minor points, this paper will make a great contribution to our understanding of neuro-behavioral parameters of disobedience in civilians vs. military personnel.

Reviewers' comments:

Reviewer's Responses to Questions

**Comments to the Author**

1. If the authors have adequately addressed your comments raised in a previous round of review and you feel that this manuscript is now acceptable for publication, you may indicate that here to bypass the “Comments to the Author” section, enter your conflict of interest statement in the “Confidential to Editor” section, and submit your "Accept" recommendation.

Reviewer #1: (No Response)

2. Is the manuscript technically sound, and do the data support the conclusions?

Reviewer #1: Yes

3. Has the statistical analysis been performed appropriately and rigorously? 

Reviewer #1: (No Response)

4. Have the authors made all data underlying the findings in their manuscript fully available?

Reviewer #1: Yes

5. Is the manuscript presented in an intelligible fashion and written in standard English?

Reviewer #1: Yes

6. Review Comments to the Author

Reviewer #1: Thank you to the authors for addressing the remaining points!

I was able to easily find the data. Is the R code provided as well? Maybe double-check whether this is necessary, too.

I recommend checking the manuscript for language errors.

Other than these two easily to address very minor points, this paper will make a great contribution to our understanding of neuro-behavioral parameters of disobedience in civilians vs. military personnel.

7. PLOS authors have the option to publish the peer review history of their article (what does this mean? ). If published, this will include your full peer review and any attached files.

**Do you want your identity to be public for this peer review?** For information about this choice, including consent withdrawal, please see our Privacy Policy .

Reviewer #1: No

---

## [Editor Report · Acceptance letter]

PONE-D-25-03927R2

PLOS ONE

Dear Dr. Tricoche,

I'm pleased to inform you that your manuscript has been deemed suitable for publication in PLOS ONE. Congratulations! Your manuscript is now being handed over to our production team.

Kind regards,

on behalf of

Dr. Rei Akaishi

Academic Editor

PLOS ONE